# Model Stitching by Invariance-aware Functional Latent Alignment

## Abstract

In deep learning, functional similarity evaluation quantifies the extent to which independently trained models learn similar input-output relationships. A related concept, representation compatibility, is investigated via model stitching, where an affine transformation aligns two models to solve a task. However, recent studies highlight a critical limitation: models trained on different information cues can still produce compatible representations, making them appear functionally similar Smith et al. (2025). To address this, we pose two requirements for similarity under model stitching, probing both forward and backward compatibility. To realize this, we introduce invariance-aware Functional Latent Alignment (I-FuLA), a novel model stitching setting. Experiments across convolutional and transformer architectures demonstrate that invariance-aware stitching settings provide a more meaningful measure of functional similarity, with the combination of invariance-aware stitching and FuLA (i.e., I-FuLA) emerging as the optimal setting for convolution-based models.

## 1 Introduction

Deep neural networks lie at the forefront of modern AI, driving significant advances across various domains ranging from computer vision (Krizhevsky et al., 2017) and natural language processing (Mikolov et al., 2013) to healthcare (Esteva et al., 2017). Their success is attributed to their ability to learn meaningful representations of the data (Bengio et al., 2012; Chowers & Weiss, 2023) which capture their abstract relationships (Doimo et al., 2020; Ziyin et al., 2024). Understanding the emergence of these representations constitutes an important problem (Kornblith et al., 2019) from both scientific and applied perspectives (Ding et al., 2021). One approach to gaining insight into learned representations is to compare the similarity of internal representations of different models. In the literature, such similarities generally fall into two categories: *representational* or *functional similarity* (Ciernik et al., 2025). In this study, we focus on the latter and formulate a functional similarity metric that quantifies the degree to which independently trained neural networks generate input-output relations following similar transitions of internal representations.

Representational similarity metrics measure correlation between geometric structures in the embedding spaces of the networks (Kornblith et al., 2019; Morcos et al., 2018; Raghu et al., 2017). Although widely used (Ciernik et al., 2025; Masarczyk et al., 2023; Mirzadeh et al., 2020; Nguyen et al., 2020), these metrics have been criticized for overestimating similarity due to spurious feature correlations (Jones et al., 2022), and for being agnostic to functional behavior (Davari et al., 2022) and invariance properties (Nanda et al., 2022) of the networks.

On the contrary, functional similarity metrics focus on the input-output behavior of the networks. Depending on the exact formulation, different notions of similarity are captured. For example, the hard (Geirhos et al., 2020) and soft (Goel et al., 2025) *error consistency* measure the extent to which identical inputs are assigned to similar outputs by different networks. On the other hand, model stitching (Csiszárik et al., 2021; Bansal et al., 2021) formulates functional similarity as embedding compatibility, i.e., whether the representations of one model can be used by another one such that a functional property of interest (e.g., classification accuracy) is maintained. In practice, this is realized by dividing each network into two parts, where the first part of one network, the *front model*, is plugged into the second part of another, the *end model*, through a trainable affine transformation

that aligns the two. The similarity is then defined as the functional property achieved by the stitched composition.

Notably, model stitching has been used to demonstrate that independently trained networks (i.e., different initialization, objective etc.) share compatible internals (Balogh & Jelasity, 2023; Bansal et al., 2021; Csiszárik et al., 2021). Recent works, however, questioned the reliability of this conclusion under the commonly used stitching setting, task loss matching (TLM) (Balogh & Jelasity, 2025; Smith et al., 2025; Hernandez et al., 2022). For example, Smith et al. (2025) demonstrated stitching compatibility between networks trained to utilize different visual cues, when solving the same task, and argued that high similarity in these cases is "misleading", a view we also share. In line with this view, we argue that a *meaningful* notion of functional similarity should reflect the extent to which two models rely on similar input patterns (i.e., operate on comparable information when solving their respective tasks).

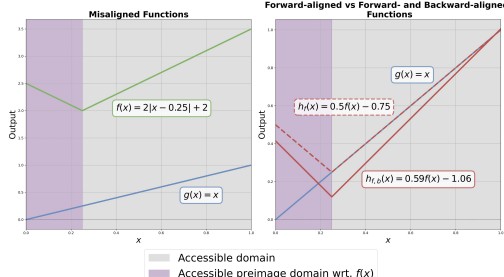

Figure 1: A toy example on function alignment under L2 minimization. When two functions $f(x)$ and $g(x)$ are aligned through a linear transformation $h(x) = W \cdot f(x) + b$ only on the accessible domain (forward-aligned), they appear perfectly similar (see $h_f(x)$). However, considering the preimage domain (forward- and backward-aligned) reveals functional misalignment even within the accessible domain (see $h_{f,b}(x)$).

In traditional model stitching under TLM, **task-level forward compatibility** is interpreted as similarity. In the other extreme, when minimizing the representation distance only at the stitch-level, that is model stitching under direct matching (DM), **stitch-level forward incompatibility** is interpreted as dissimilarity. Ultimately, relying solely on forward compatibility fails to capture the extent to which invariant properties are shared among the stitched models i.e., whether they are **backward compatible**. Without considering backward compatibility, functional discrepancies between models may be obscured, leading to spurious assessments of functional similarity. We highlight the relevance of incorporating backward comparability through a toy functional alignment example in Fig 1. Building on this, Fig 2 formalizes our conceptual arguments on why previously explored model stitching approaches are unsuitable for functional similarity evaluation. Finally, we pose two requirements that the networks need to meet to be considered functionally similar under model stitching.

**Requirement. A (Latent-level forward compatibility):** For each input, the standalone end model, and the end model within the stitched composition process the internal representations following the stitching in a similar manner, i.e., the representations undergo similar transitions throughout the layers.

**Requirement. B (Backward compatibility):** For each pair of inputs that the front model represents identically at the stitch-level, the end model processes the internal representations following the stitching in a similar manner.

Towards model stitching that better fits the needs of functional similarity evaluation, we strive for developing a setting that accounts for both the forward and backward notions of compatibility. To accommodate for **Req. A**, we propose Functional Latent Alignment (FuLA) as a novel stitching objective, in which the affine transformation minimizes not only the feature representations at the stitch-level (i.e., DM) but also those at the layers following it, up until the penultimate layer (i.e., excluding the output). To probe **Req. B**, we use Identically Represented Inputs (IRIs) Nanda et al. (2022) to establish the stitching alignment under the invariance class of inputs induced by the front model at the stitch-level. Our contributions are as follows:

- We propose FuLA as a novel model stitching setting (Sec. 2.2).
- We show that existing stitching settings (including FuLA) not only fail to distinguish between models relying on different cues (Smith et al., 2025) but can also utilize previously unseen cues to improve alignment (Sec. 3.2).
- We show that utilizing the invariance induced by the front model (i.e., training on IRIs) to establish the alignment (invariance-aware stitching), mitigates the above shortcoming (see Secs. 3.1 and 3.2).

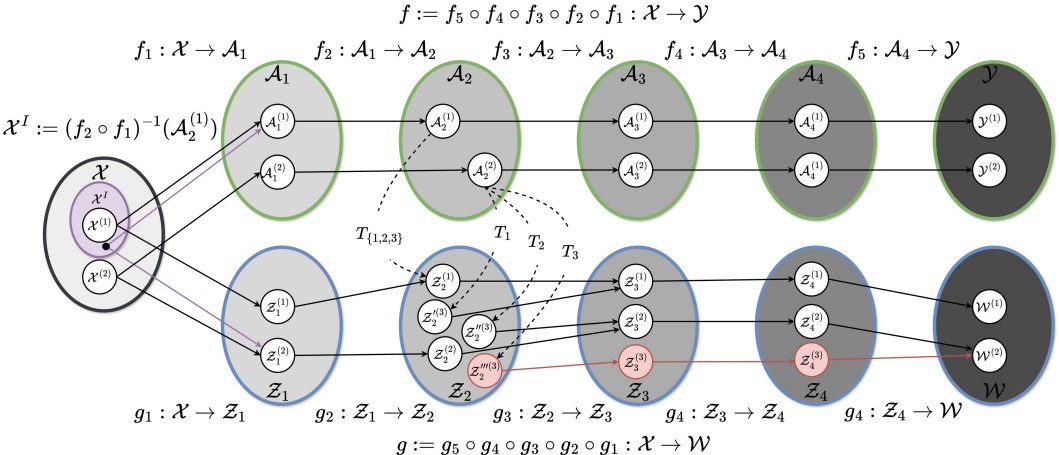

Figure 2: A motivating example of functional alignment in model stitching (best viewed in color). We are interested in evaluating the functional similarity of the compositions $f : \mathcal{X} \to \mathcal{Y}$ (front model) and $g : \mathcal{X} \to \mathcal{W}$ (end model) at the $2^{\text{nd}}$ layer. Let $T_1$, $T_2$ and $T_3$ be three different transformations, mapping domain $\mathcal{A}_2$ to $\mathcal{Z}_2$, that only differ in how they map the subdomain $\mathcal{A}_2^{(2)}$ into $\mathcal{Z}_2'^{(3)}$, $\mathcal{Z}_2''^{(3)}$ and $\mathcal{Z}_2'''^{(3)}$ respectively, all three positioned equidistantly from $\mathcal{Z}_2^{(2)}$. A systematic divergence from the intermediate subdomains utilized by the standalone end model constitutes an unbounded expansion of its internal states. Under the DM, all three transformations are treated as equivalent. In contrast, TLM regards $T_2$ and $T_3$ as equally performant, as they both preserve the input-output relationship of the end model, despite $T_3$ inducing an unbounded expansion of internal states. Importantly, although the stitched compositions realized by $T_2$ and $T_3$ establish optimal (forward) alignment at the task-level, they fail to reveal the (backward) misalignment with respect to $\mathcal{X}^I$, the preimage of $\mathcal{A}_2^{(1)}$.

- In light of the above, we revisit model stitching between robust and non-robust networks and show that they are not as functionally similar as suggested by the literature (Sec. 3.3).

- We conclude that invariance-aware model stitching provides more meaningful functional similarity evaluation for the considered convolution-based models (ResNet-18 and VGG-16) and the transformer-based model (ViT-Tiny), with FuLA and soft label matching being the respective optimal choices, despite task loss matching being dominantly used in the literature.

## 2 MODEL STITCHING

In this section, we introduce the notation and provide an overview of the components that form the basis of our experimental setup and introduce FuLA as a model stitching setting.

### 2.1 NOTATION

We closely follow the notation used by Balogh & Jelasity (2025). Let $f_w : \mathcal{X} \to \mathcal{Y}$ be a feedforward neural network with $m$ layers, parameterized by $w$, which we omit for brevity when clear from the context. We can write $f$ as the composition $f := f_m \circ \cdots \circ f_1$, where each $f_i : \mathcal{A}_{i-1} \to \mathcal{A}_i$ maps the activation space of the $(i-1)^{\text{th}}$ to that of the $i^{\text{th}}$, with $\mathcal{A}_0 = \mathcal{X}$. For $i \geq 1$ and $r \leq m$, we define the partial compositions $f_{[i:r]} := f_r \circ \cdots \circ f_i$ as well as $f_{>i} := f_{[i+1:m]}$ and $f_{\leq i} := f_{[1:i]}$ such that $f = f_{>i} \circ f_{\leq i}$. Similarly, we assume $g_\phi : \mathcal{X} \to \mathcal{W}$ with $g := g_k \circ \cdots \circ g_1$ and $g_j : \mathcal{Z}_{j-1} \to \mathcal{Z}_j$.

Model stitching for functional similarity evaluation aims at quantifying the extent to which $g_{>j}$ can maintain the functionality of $g$ when given as input the $\mathcal{A}_i$ with $i \in [m]$ and $j \in [k]$. Formally, the stitched model is constructed as $h_\theta = g_{>j} \circ T_\theta \circ f_{\leq i}$. Here, $T_\theta : \mathcal{A}_i \to \mathcal{Z}_j$ is referred to as the stitching layer, and we refer to $f$ and $g$ as front and end models, respectively. In practice, the optimal $T_\theta$, under a relevant optimality criterion, is selected from a suitable family of transformations $\mathcal{S}$,

while $g$ and $f$ are kept fixed. Given the nature of the problem, the transformation $T_\theta \in \mathcal{S}$ needs to be sufficiently flexible to allow non-trivial mappings while ensuring that the capacity of the $h$ does not exceed that of the combined partial networks $g_{>j}$ and $f_{\leq i}$. The family $\mathcal{S}$ of affine transformations is considered in the literature to satisfy these requirements (Balogh & Jelasity, 2025).

In our work, we considered functional similarity evaluation of image classification networks, the predominant use case in the relevant literature. We assume access to two classification datasets, $\mathcal{D}_{\text{train}} = \{(x_s, y_s)\}_{s=1}^n$ and $\mathcal{D}_{\text{test}} = \{(x_s, y_s)\}_{s=1}^c$, where $x_s$ denotes an image and $y_s$ its corresponding label in one-hot vector format. The transformation $T_\theta$ is optimized using $\mathcal{D}_{\text{train}}$, while the functional similarity is evaluated on the $\mathcal{D}_{\text{test}}$.

The optimization objectives previously explored in the literature are the TLM (Bansal et al., 2021; Csiszárik et al., 2021), the soft label matching (SLM) (Csiszárik et al., 2021) [1] and the DM (Balogh & Jelasity, 2025) defined as:

$$\mathcal{L}_{\text{TLM}}: \quad \arg\min_\theta \mathbb{E}_{p(x,y)}\Big[\mathcal{L}_{\text{CE}}\big((g_{>j} \circ T_\theta \circ f_{\leq i})(x), y\big)\Big], \tag{1}$$

$$\mathcal{L}_{\text{SLM}}: \quad \arg\min_\theta \mathbb{E}_{p(x)}\Big[\mathcal{L}_{\text{CE}}\big((g_{>j} \circ T_\theta \circ f_{\leq i})(x), g(x)\big)\Big], \tag{2}$$

$$\mathcal{L}_{\text{DM}}: \quad \arg\min_\theta \mathbb{E}_{p(x)}\Big[\big\|(T_\theta \circ f_{\leq i})(x) - g_{\leq j}(x)\big\|_F\Big], \tag{3}$$

with $\mathcal{L}_{\text{CE}} : \mathcal{Y} \times \mathcal{Y} \to \mathbb{R}$ denoting the cross-entropy loss and $\|\ \|_F$ the Frobenius norm.

## 2.2 FUNCTIONAL LATENT ALIGNMENT (FuLA)

To this end we hypothesize that the existing stitching objectives violate **Req. A**, which is related to forward compatibility. In particular, TLM and SLM may be overly **forgiving** settings by exploiting irregularities in the models' decision process to achieve optimal hard- or soft-label matching performance, respectively. Conversely, the DM setting can be overly **penalizing**, as it does not take into account the subspaces relevant to the layers (i.e., the flow of activations) following the stitching. Based on these, we propose FuLA where the stitched composition is explicitly optimized to mimic the internal processes from the stitching layer up until the penultimate layer of the standalone end model. Formally the $\mathcal{L}_{\text{FuLA}}$ objective writes as:

$$\arg\min_\theta \mathbb{E}_{p(x)}\left[ C_j \underbrace{\frac{\big\|(T_\theta \circ f_{\leq i})(x) - g_{\leq j}(x)\big\|_F}{\|g_{\leq j}(x)\|_F}}_{\mathcal{L}_{\text{Hint}}^j} + \sum_{l=j+1}^{k-1} C_l \underbrace{\frac{\big\|g_{[j+1,l]}\big((T_\theta \circ f_{\leq i})(x)\big) - g_{\leq l}(x)\big\|_F}{\|g_{\leq l}(x)\|_F}}_{\mathcal{L}_{\text{Hint}}^l} \right], \tag{4}$$

where $C \in \{c \in \mathbb{R}^{k-j} : c_l \geq 0, \sum_{l=j}^{k-j} c_l = 1\}$ defines a weighted average that controls the relative contribution of each term. We use uniform weighting as the default option. Additionally, we divide by the target's norm to account for potential differences in scale between feature activations at different depths. We refer to these terms as Hints (Romero et al., 2014), denoted as $\mathcal{L}_{\text{Hint}}^t$, where $t$ indicates the depth. Note that for $t = j$, the structural Hint $\mathcal{L}_{\text{Hint}}^j$ (up to scale) corresponds to the direct matching objective defined in Eq. (3), while the functional Hints $\mathcal{L}_{\text{Hint}}^l$ for $t = l \geq j + 1$ realize function alignment (as per **Req. A**). Ultimately, the transformation $T_\theta$ is trained to map the $\mathcal{A}_i$ into $\mathcal{Z}_j$ such that its output is interpreted by $g_{>j}$ similarly to its native input originating from $g_{\leq j}$. For a visual overview of the model stitching objectives and their relation to FuLA, we refer readers to Fig. 8 in the supplementary material.

---

[1] Csiszárik et al. (2021) use the term TLM interchangeably to refer to both TLM and SLM, and argue that these two settings lead to highly correlated results. However, we found that TLM and SLM can behave differently under certain key stitching configurations, and therefore, we distinguish them.

### 2.3 TRAINING DATA IN MODEL STITCHING

In this work, we leverage the concepts of *Identically Represented Inputs (IRIs)* (Nanda et al., 2022) and *adversarial training (AT)* (Madry et al., 2017) both of which involve altering the training data used to establish the alignment in model stitching.

#### 2.3.1 GENERATING IDENTICALLY REPRESENTED INPUTS

Given an input $x$ and a feature extractor $f_{\leq i}$, for a small enough tolerance $\rho > 0$ we define the set:

$$\text{IRIs}_{\text{relax}}(x; f_{\leq i}) = \{\, x' \mid \underbrace{\frac{||f_{\leq i}(x') - f_{\leq i}(x)||_F}{||f_{\leq i}(x)||_F}}_{\mathcal{L}^i_{\text{Hint}}} \leq \rho \},\tag{5}$$

which consists of all inputs $x'$ that yield *almost* the same representation as $x$ under $f_{\leq i}$. For $\rho = 0$, this reduces to the exact representation-invariant set $\text{IRIs}(x; f_{\leq i})$, which is hard to attain in practice. Following (Nanda et al., 2022), we obtain $\text{IRIs}_{\text{relax}}(x; f_{\leq i})$ by optimizing trainable inputs $x'$, initialized by uniform random noise, to minimize the IRIs Hint $\mathcal{L}^i_{\text{Hint}}$. We then construct the dataset $\mathcal{D}^{\text{IRIs}}_{\text{train}} = \{(x'_s, y_s)\}^n_{s=1}$ by replacing each input $x_s$ from $\mathcal{D}_{\text{train}}$ with a corresponding sample $x'_s \in \text{IRIs}_{\text{relax}}(x_s; f_{\leq i})$, while we consider $f$ as the front model which we stitch into the end model $g$ at stitch-level $i$.

By construction, corresponding pairs between $\mathcal{D}^{\text{IRIs}}_{\text{train}}$ and $\mathcal{D}_{\text{train}}$ are *almost* indistinguishable to $f$ at the stitch-level. We assume that optimizing the stitching layer on $\mathcal{D}_{\text{train}}$ achieves the *unique* optimal forward compatibility on $\mathcal{D}_{\text{test}}$. Consequently, any deviation from this optimum observed when training on $\mathcal{D}^{\text{IRIs}}_{\text{train}}$ constitutes a violation of **Req. B**. In practice, to probe **Req. B**, we establish the stitching alignment on $\mathcal{D}^{\text{IRIs}}_{\text{train}}$ while evaluating the functional similarity on $\mathcal{D}_{\text{test}}$.

#### 2.3.2 ADVERSARIAL TRAINING

AT is widely regarded as the most effective method for improving model robustness against worst-case perturbations, referred to as adversarial examples (Goodfellow et al., 2014; Li et al., 2022). In this case, the training objective for the neural network $f_w$ becomes:

$$\arg \min_w \mathbb{E}_{p(x,y)} \left[ (1 - \alpha)\mathcal{L}_{\text{CE}}(f_w(x), y) + \alpha \max_{\delta \in \mathcal{B}(x,\epsilon)} \mathcal{L}_{\text{CE}}(f_w(x + \delta), y) \right],\tag{6}$$

where $\alpha \in [0, 1]$ controls the proportion of adversarial samples and $\mathcal{B}(x, \epsilon)$ denotes the $\ell_\infty$-ball of radius $\epsilon$. In this work, we revisit the functional similarity between robust and non-robust networks (Balogh & Jelasity, 2023) where we train robust models via AT to serve as the front and/or end models. Additionally, we leverage AT during the optimization of the stitching composition $h_\theta$ (i.e., replacing $f_w$ with $h_\theta$ in Eq. (6)) to evaluate functional alignment with respect to adversarial samples.

## 3 ANALYZING FUNCTIONAL SIMILARITY VIA MODEL STITCHING

To this end, we hypothesize that accounting for both **Reqs. A** and **B** provides a more meaningful model stitching setting for functional similarity evaluation. We investigate the effect of **Req. A** by comparing the stitching objectives presented in Sec. 2. In relation to **Req. B**, we compare the effects of optimizing the stitching layer on either on $\mathcal{D}_{\text{train}}$ or $\mathcal{D}^{\text{IRIs}}_{\text{train}}$, while reporting similarity on $\mathcal{D}_{\text{test}}$ in both cases. We refer to the settings trained on $\mathcal{D}^{\text{IRIs}}_{\text{train}}$ as invariance-aware and denote them with the prefix I- (e.g., I-FuLA). Note that probing for Req. B requires incorporating perception from the end model when optimizing the stitching layer. Although TLM can not be turned into invariance-aware, as it does not incorporate perception from the end model, we slightly abuse the notation and evaluate I-TLM as a reference for the quality of $\mathcal{D}^{\text{IRIs}}_{\text{train}}$[2].

---

[2]In principle, model stitching under TLM should behave identically when optimized under on either the exact $\mathcal{D}^{\text{IRIs}}_{\text{train}}$ or $\mathcal{D}_{\text{train}}$.

In the absence of objective measures of meaningfulness in this context, we rely on sanity checks grounded on intuitive expectations (see Secs 3.1 and 3.2), which have formed the basis of critiques in recent literature.

**Stitching layer:** The stitching layer $T_\theta$ is modeled as a $1 \times 1$ convolutional layer including a bias term, while keeping the front and end models frozen, following Csiszárik et al. (2021). We initialize $T_\theta$ by linearizing the DM objective in Eq. (3) and solving it using the Moore-Penrose pseudoinverse with 500 training samples (Balogh & Jelasity, 2025; Csiszárik et al., 2021).

**Experimental setting:** We consider image classification networks and conduct our experiments using CIFAR-10 (Krizhevsky et al., 2009) and LS-ImageNet-10, a low-resolution version of ImageNet (Deng et al., 2009) from which we randomly sampled 10 classes, and modified variants of these (e.g., grayscale versions). Throughout the paper, we use the ResNet-18 (He et al., 2016) architecture as the trained front and end models. We perform stitching at the first convolutional layer and at each residual block. For each stitching configuration, we report the functional similarity averaged across three random initializations.

**The stitching plot:** When stitching corresponding layers (i.e., $i = j$), we use the stitching plot (Balogh & Jelasity, 2023), where the y-axis represents a relevant functional property and the x-axis denotes the depth of the front layer composition. The first and last points in the x-axis correspond to the baseline end and front models respectively. Previous works use top-1 (robust) accuracy of the stitched composition on the end model's task to infer functional similarity. However, we argue that top-1 measures can overlook finer aspects of similarity (e.g., per-sample predictions). Instead, we utilize the *agreement probability* (Goel et al., 2025) with the end model's predictions, capturing similarity of the full class probability distributions in a sample-wise manner.

By design, the stitching plot between a front and an end model at $M$ corresponding layers begins at $(0, 100\%)$ and ends at $(M + 1, Q)$, where $Q$ is the agreement probability achieved by the standalone front model relative to the end model. For example, a data point at $(3, 95\%)$ suggests that replacing the first three layers of the end model with those of the front results in a stitched composition with $95\%$ agreement probability, with respect to the standalone end model's predictions. Intuitively, we expect the stitching plots of similar networks to resemble a smooth transition from $100\%$ to $Q$ agreement probability.

### 3.1 FUNCTIONAL SIMILARITY ACROSS INFORMATION VARIANTS

Smith et al. (2025) showed that models trained to rely on different cues (e.g., color, bias etc.) are functionally similar under TLM, and argued that this is an undesirable property of functional similarity evaluation. We revisit this observation by stitching front models trained on different CIFAR-10 variants into **the same** end model trained on RGB CIFAR-10.

In particular, we stitching into the end model the following front models: (1) the end model itself (identity), (2) a different random initialization of the end model trained on RGB CIFAR-10 (cross-seed), (3) a model trained on grayscale CIFAR-10 (cross-data), and (4) a model trained on RGB CIFAR-10 with injected class-correlated pixel shortcuts (cross-bias) (see Fig. 3). For model stitching involv-

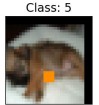 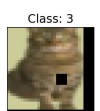 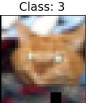
Class: 5  Class: 3  Class: 3

Figure 3: Pixel shortcuts.

ing (1)–(3), we optimize the stitching layer and report the similarity on the RGB CIFAR-10 train ($\mathcal{D}_\text{train}$) and test splits ($\mathcal{D}_\text{test}$). For (4) we use the pixel-injected versions of these splits, ensuring that both the front and end models have access to their predictive visual cues[3].

In Fig. 4 we provide the results for all stitching settings and observe that learning the alignment on $\mathcal{D}_\text{train}$ (blue curves) leads to similar agreement interpolation trends across all configurations, indicating high functional similarity. When stitching (4), DM constitutes an exception where a dip occurs at the mid layers, but high similarity is recovered in the deeper layers. These results suggest that, under the currently available notions of forward compatibility, including FuLA which accounts for **Req. A**, model stitching is not sufficient to differentiate among model compositions relying on different information cues.

---

[3]The models trained on pixel-injected RGB CIFAR-10 performs only marginally above random chance when evaluated on the RGB CIFAR-10, whereas the model trained on RGB CIFAR-10 perform comparably on both RGB CIFAR-10 and its pixel-injected variations.

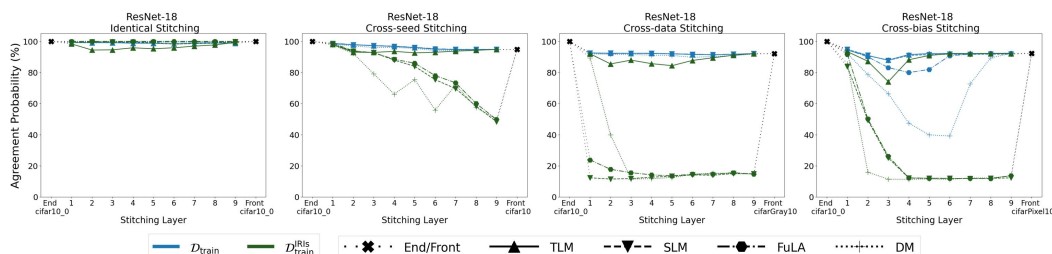

Figure 4: Stitching between models relying on different information. Note that the settings probing only for forward compatibility display similar qualitative trends, where generally the functional similarity is interpolated between the standalone end and front models. Probing for backward compatibility (i.e., Req. B) leads to significantly different, and fundamentally more intuitive, functional similarity trajectories.

Next, we establish the alignment on the $\mathcal{D}_{train}^{IRIs}$ (green curves), effectively probing **Req. B**. As expected, the I-TLM closely follows the blue curves. On the contrary, the I-SLM, I-FuLA and I-DM, display a declining trend for (2) supporting the multi-view hypothesis (Allen-Zhu & Li, 2023), and a quickly decaying similarity for (3) and (4). In that sense (1) appears to be more functionally similar to (2) compared to (3) and (4), an observation that aligns with our intuition. Interestingly, independently trained models are not as functionally similar as initially suggested by forward compatibility in isolation. Overall, the DM-based objectives tend to achieve the lowest degrees of similarity. Based on these, the I-SLM and I-FuLA emerge as relevant model stitching settings that are sensitive to information differences and therefore can provide a more meaningful notion of functional similarity.

To rule out the possibility that the effect observed for the invariance-aware objectives is merely a byproduct of a poorly constructed $\mathcal{D}_{train}^{IRIs}$, we performed a series of relevant sanity tests (see Sec. 9).

## 3.2 FUNCTIONAL SIMILARITY UNDER UNSEEN PREDICTIVE INFORMATION

In Sec. 3.1, we showed that forward compatibility can be attained under several stitching objectives, beyond TLM. We now investigate whether model stitching can also leverage previously unseen information to improve compatibility.

To this end, we introduce an additional front model: (5), a model trained on LS-ImageNet-10, which is stitched into the same RGB CIFAR-10 end model used earlier (cross-task). To establish and evaluate the alignment, we consider three variants of RGB CIFAR-10: (i) pixel-injected data at fixed location for each class, (ii) pixel-injected data and (iii) data from (ii) where the pixel patterns are replaced with random noise (i.e., negating the shortcuts). Among these, (i) provides the most available and predictive cues followed by (ii), whereas (iii) provides none. Note that neither the front nor the end models have previously seen these class-correlated patterns and therefore any utilization of those originate from the stitching layers exploiting them.

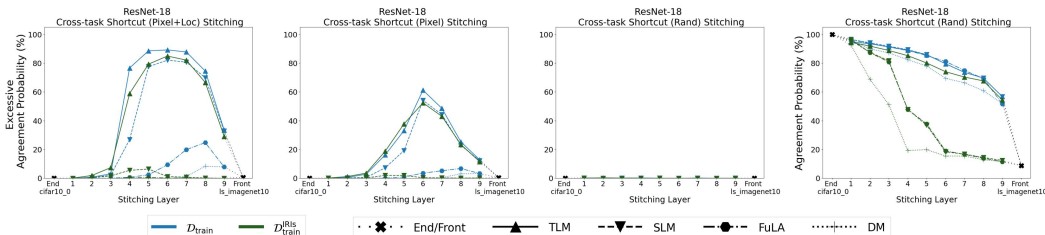

Figure 5: Stitching under shortcuts of varying availability. Note that the settings probing only for forward compatibility are susceptible to relying on previously, but predictive, cues to establish alignment, as indicated by the elevated excessive agreement probability. The effect decays as we decrease the availability of the shortcuts. Ultimately, probing for backward compatibility (i.e., Req. B) significantly mitigates the effect.

To quantify the extent to which the stitched composition utilizes the pixel patterns, in Fig 5 we report the excessive agreement probability, that is, the difference between the agreement probability on the pixel-injected test split and that on a variant where the pixel patterns are shifted among classes. Intuitively, a larger difference indicates that the stitched composition relies more heavily on the pixel cues presented during training.

When it comes to the alignment found on $\mathcal{D}_{\text{train}}$, we observe that all settings are susceptible to learning shortcuts, with the effect diminishing as we progress from (i)–(iii), irrespective of the stitching objective. This finding suggests that the stitched composition can achieve (excessive) forward compatibility by exploiting previously unseen patterns. This casts further doubt on forward compatibility as a measure of functional similarity, since learning to rely on previously unused cues is undesirable in this context Bansal et al. (2021). Crucially, showing that there exists any configuration under which model stitching systematically engages in learning is sufficient evidence that such behavior can arise, thereby calling into question the general reliability of model stitching for functional similarity evaluation. In our experiments, we observed more pronounced instances of this behavior in cross-task stitching configurations and therefore adopted this setting as our primary focus. Nevertheless, the effect also appeared when stitching models trained on the same dataset, though less consistently and to a much smaller extent (see Fig. 15 in the supplementary material).

Fortunately, the behavior shifts considerably when incorporating the backward compatibility requirement (**Req. B**). In this case, we observe that excessive agreement drops close to zero for I-FuLA and I-DM, essentially achieving similar behavior to (iii) even when presented with highly available and predictive patterns such as (i). Although I-SLM still engages in some shortcut learning, it does so to a significantly lesser extent compared to SLM. Finally, when evaluating the agreement probability on (iii), we find that DM-based underperforms the respective SLM and FuLA counterparts, with I-FuLA being the overall optimal in these configurations.

## 3.3 Functional Similarity between Robust and non-Robust models

Perhaps surprisingly, Balogh & Jelasity (2023) showed that robust and non-robust networks are functionally similar under model stitching by TLM. As we saw earlier, forward compatibility (e.g., TLM) not only fails to distinguish between models that rely on different cues to solve a task, but can also exploit cues unintended by either the front or end models. This observation motivates revisiting the question of functional similarity between robust and non-robust networks. Additionally, the multi-objective nature of the robust to non-robust configurations makes it a relevant test bed for evaluating the interplay of **Reqs. A** and **B** in functional similarity evaluation.

We consider the model (1) used earlier, and (6) a model trained on RGB CIFAR-10 using AT of Eq.(6) with $\alpha = 0.5$. Similar to (Balogh & Jelasity, 2023), we use (1) and (6) to realize all possible model stitching configurations. In this case, the similarity with respect to both clean and adversarial samples is of interest. However, this similarity could in principle be influenced by the stitching objective used to establish the alignment. To account for this, we optimize the stitching layer and evaluate similarity on clean and adversarial samples independently, thereby probing each sample type in isolation.

We established earlier that TLM is a suboptimal setting for similarity evaluation, even when used on IRIs data and therefore we omitted from this analysis. To ease inter-configuration comparison when the standalone front models achieve significantly different agreement probability, we compute the relative Area under Agreement (rAuA) defined as the average agreement probability achieved when stitching divided by that of the standalone front and end models. The rAuA measures the deviation from the smoothly declining agreement curve, where values closer to zero indicates sharp drops in probability agreement and therefore functional incompatibility.

In Fig.6 we provide the results obtained when stitching between various combination of robust and non-robust models. Under regular model stitching, DM and FuLA display a smooth similarity transition during stitching, indicating compatibility for both clean and adversarial samples, for all configurations. On the other hand, SLM displays an erratic behavior when there is a mismatch between the end model and the stitching objective (see Fig. 6, second row, left two panels).

When evaluating the invariance-aware settings, we observe that the similarity depends heavily on the specific configurations. For example under I-FuLA robust end models are more similar, with respect

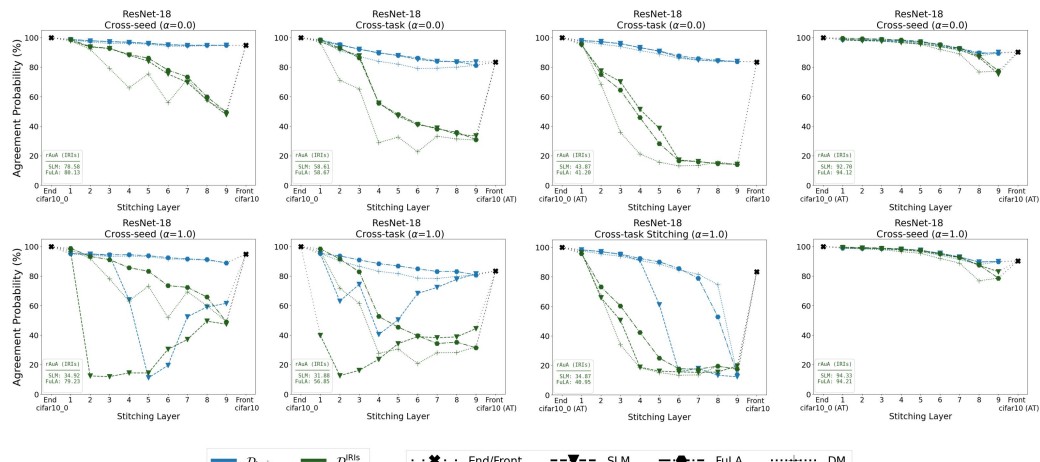

Figure 6: Stitching between robust and non-robust models. The "clean(adversarial)" label indicates that the stitching layer is optimized and evaluated using only clean(adversarial) samples. Note that the settings probing only for forward compatibility indicate the robust and non-robust models are functionally similar with respect to regular samples (i.e., first row). However, probing for backward compatibility (i.e., Req. B) reveals functional incompatibilities previously overlooked.

to clean inputs, to robust front models compared to non-robust front models, a property missed by FuLA.

Interestingly, under the invariance-aware stitching it is evident that pairs of robust models are more similar than pairs of non-robust models. Analogous observations were made in (Nanda et al., 2022; Jones et al., 2022), but from a representational similarity standpoint. Across most configurations, DM-based stitching achieves the lowest similarity. Based on these, I-FuLA emerges as the most well-behaved setting for evaluating functional similarity between robust vs non-robust networks.

## 3.4 FURTHER RESULTS AND DISCUSSION

We repeated the experiments on the VGG-16 (Simonyan & Zisserman, 2014), ViT-Tiny and ViT-Base (Dosovitskiy et al., 2020) architectures and extend our analysis to ResNet-18 and ViT-Base architectures trained on ImageNet subsets (e.g., ImageNet-100 (Sarıyıldız et al., 2023)). Following the same line of reasoning used for ResNet-18, we conclude that invariance-aware stitching, motivated by **Req. B**, is relevant across all considered architectures, as it yields more meaningful functional similarity measurements. Additionally, we found **Req. A** to be applicable to both ResNet-18,VGG-16 and Vit-Base, with I-FuLA emerging as the optimal setting. Although I-SLM was identified as the optimal setting for ViT-Tiny, **Req. A** remains partially relevant, as I-FuLA generally outperformed I-DM while consistently passing all relevant sanity checks. Additionally, we compare I-FuLA (and FuLA) with CKA and I-CKA (Nanda et al., 2022), showing that I-FuLA provides unique insights beyond what standard representation similarity metrics capture, even when accounting for shared invariances (i.e., I-CKA) (see Sec. H).

**A note on convergent representations:** At first glance, our findings appear to contradict a growing body of literature on convergent representations. For example, Huh et al. (2024) showed that representations of progressively more capable models converge toward an aligned representational structure. Moreover, the broad success of model stitching applications further reinforces this view (Balogh & Jelasity, 2023; Moschella et al., 2022). On the other hand, there are works, including ours, that point the other way. For example, the notion of convergent representations can be incompatible with the multiview hypothesis (Allen-Zhu & Li, 2023). To this end, we argue that both representation convergence and functional divergence can co-exist. That is, it is possible that representations converge to a structurally similar latent space while the models that generate them realize fundamentally different functions.

## 4 RELATED WORK

**Model Stitching:** Model stitching (Lenc & Vedaldi, 2015) has been used to connect independent network components from the model zoo (Pan et al., 2023; 2024; Yang et al., 2022), allowing for networks with controllable performance-efficiency trade-offs. Moreover, it was shown that models can be successfully stitched either by direct transformation in the latent space (Lähner & Moeller, 2024; Maiorca et al., 2023) or in zero-shot fashion by learning representations in relative space (Cannistraci et al., 2023b;a; Moschella et al., 2022). In contrast to these works, we employ model stitching as a means for functional similarity evaluation (Bansal et al., 2021; Csiszárik et al., 2021), where task performance is required to correlate with similarity.

**Feature distillation:** Feature-based distillation was introduced by Romero et al. (2014), where teacher features guide the student via L2 minimization. Notable extensions include attention-based distillation (Zagoruyko & Komodakis, 2016) and inter-channel correlation transfer (Liu et al., 2021), both applied at manually chosen layers. Ji et al. (2021) and Chen et al. (2021) avoid manual layer linking by learning connections via attention. While FuLA also minimizes L2 feature distance, it does so through frozen layer compositions of the end model, achieving alignment in the functional sense.

**Function consistency:** Function consistency has been used in KD (Liu et al., 2023; Yang et al., 2021) to help students better replicate teacher behavior, improving performance. Yang et al. (2022) applied it to group equivalent blocks in the model zoo, which were then reassembled under resource constraints. In contrast, FuLA leverages function consistency to align models for measuring functional similarity.

## 5 CONCLUSION

In this work, we examine the problem of functional similarity evaluation through model stitching. While previous findings suggest that model stitching cannot reliably distinguish between models trained on different information, we further demonstrate that the stitched composition may exploit predictive but unintended cues, unused by both the front and the end model, to achieve alignment. To account for this, we pose two requirements, based on which we reformulate model stitching to probe for both forward and backward compatibility. Alongside this, we revisit stitching between robust and non-robust models, yielding insights that differ from the previous literature, which probed only forward compatibility. Furthermore, we show that incorporating backward compatibility leads to more meaningful functional similarity measurements in both convolution- and transformer-based architectures, while the combination of backward and forward latent-level compatibility emerges as the optimal setting for convolution-based architectures. Finally, our work highlights the relevance of approaching similarity evaluation from multiple perspectives, thereby improving our understanding of neural network internal processes.

**Limitations:** An inherent limitation of this work is the absence of an "oracle" notion of functional similarity. As a result, definitive conclusions about which settings provide the most faithful reflection of similarity remain out of reach. Furthermore, our study is limited to image classification models. Exploring the interplay of forward and backward compatibility in model stitching for other tasks within the same modality and across different modalities is a promising direction for future work.

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

## A EXPERIMENTAL SETUP

In this section, we cover the implementation details needed for reproducing our results.

### A.1 BASELINE TRAINING

In this study, we experimented with convolution-based architectures, namely ResNet18 and VGG16, as well as a vision-based ViT-Tiny and Vit-Base. For the low-resolution settings, we used CIFAR-10 and a low-resolution version of ImageNet, from which we randomly sampled 10 classes. We refer to this dataset as LS-ImageNet-10. For the high-resolution settings we used ImageNet-100 Sarıyıldız et al. (2023) as well as ImageNetB-100 which we created by randomly sample a set of 100 classses from ImageNet-1K not included in ImageNet-100.

**Non-robust models:** We considered ResNet-18, VGG-16 and ViT-Tiny for the low-resolution setting and ResNet-18 and ViT-base for the high-resolution one. We used identical training recipes for both low- and high-resolution settings. The ResNet-18, ViT-Tiny and ViT-base non-robust models were trained for 200 epochs using the SGD optimizer with momentum. The initial learning rate was set to 0.1, with a batch size of 128. During training, we applied random horizontal flips and random cropping with 4-pixel padding. A weight decay of $1 \times 10^{-4}$ was used, and the learning rate was reduced by a factor of 0.1 at every third of the total training epochs. The same hyperparameters were used for VGG-16, with the only exception being the learning rate, which was set to 0.01.

**Robust models:** The robust models were trained using the same hyperparameter settings as the non-robust models, with adversarial training performed according to Madry et al. (2017), using an equal mix of clean and adversarial examples ($\alpha = 0.5$). For the low-resolution setting, the adversarial examples were generated using 10 steps, with a step size of $2/255$ and $\epsilon$ set to $8/255$. Following Balogh & Jelasity (2023), we measure adversarial robustness using the "rand" configuration of AutoAttack Croce & Hein (2020). For the high-resolution settings we used a step size of $1 \cdot 4.758/255$ and $\epsilon$ set to $4 \cdot 4.758/255$.

### A.2 STITCHING TRAINING

For convolution-based architectures, we follow Balogh & Jelasity (2023); Csiszárik et al. (2021) and use $1 \times 1$ convolutional layers with bias as a stitch-layer to connect the front and end models. During training, we keep everything frozen apart from the alignment transformation. When training on $\mathcal{D}_{\text{train}}$, we allowed the update of the batch normalization statistics of the front and end models, while when training on $\mathcal{D}_{\text{train}}^{\text{IRIs}}$, we kept the batch statistics frozen too to avoid accumulating errors when invariances do not hold for all layers. We perform stitching at the first convolutional layer and at the first and last (residual) block of each layer, leading to 9 stitching locations for all architectures considered. For VGG-16, we perform stitching at the 5 layers followed by the max pooling layers.

For ViT-Tiny, we follow Balogh & Jelasity (2025) and use a linear layer, applied on the feature representation of each token, to connect the front and end models. We perform stitching at the end of each of the 12 encoder blocks and at the final cls token before the classifier. For Vit-Base, we follow the procedure used for Vit-Tiny, with the only expectation being that stitched at few encoder blocks (i.e., 6 in total). Again, we follow Balogh & Jelasity (2025) and use a linear layer, applied on the feature representation of each token, to connect the front and end models. We perform stitching at the end of each of the 12 encoder blocks and at the final cls token before the classifier.

The stitching transformations were trained for 30 epochs using the Adam Kingma (2014) optimizer with a learning rate of 0.001 and weight decay of $1 \times 10^{-4}$. During training, we used the same augmentation strategy used to train the base models. At all times, the stitching layer was initialized by DM using 500 samples for the training set.

We conducted all experiments across three randomly initialized runs. For each configuration, we used the same end model instance, while we used a different front model for each run.

**Model Stitching under Adversarial Training:** When performing stitching under adversarial training, we consider two $\alpha$ configurations, $\alpha = 0$ (clean), $\alpha = 1.0$ (adversarial). The adversarial examples were generated for the stitched model using the same hyperparameters as those used to train the robust models.

# B CONSTRUCTING $\mathcal{D}_{\text{TRAIN}}^{\text{IRIS}}$

Given a front model $f$ and stitch-level $i$, we construct the $\mathcal{D}_{\text{train}}^{\text{IRIS}}$ by sampling a data point $x'$ from the $\text{IRIS}_{\text{relax}}(x; f_{\leq i})$, for each data point x in $\mathcal{D}_{\text{train}}$.

$$\text{IRIS}_{\text{relax}}(x; f_{\leq i}) = \{ x' \mid \underbrace{\frac{||f_{\leq i}(x') - f_{\leq i}(x)||_F}{||f_{\leq i}(x)||_F}}_{\mathcal{L}_{\text{Hint}}^i} \leq \rho \}. \tag{7}$$

In practice, we generate these $x'$ by minimizing $\mathcal{L}_{\text{Hint}}^i$ using 200 steps, with a step size of 1 and $\epsilon$ set to $1/255$.

# C MODEL STITCHING UNDER SHORTCUT:

We consider various shortcut settings, namely: (i) pixel-injected data at a fixed location for each class, (ii) pixel-injected data, or (iii) data from (ii) where the pixel patterns are replaced with random noise (i.e., negating the shortcuts) (see Fig. 7).

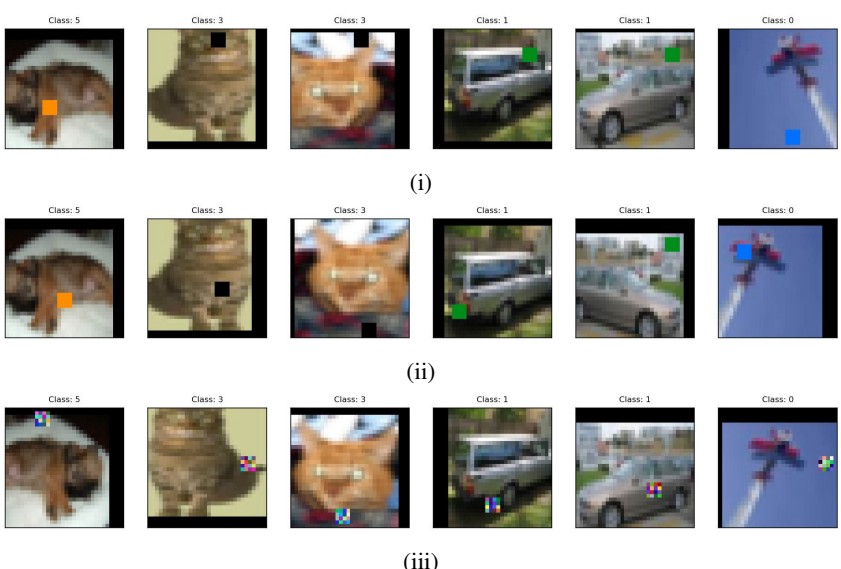

Figure 7: Stitching under shortcuts of varying availability.

# D FULA

When realizing the FuLA objective, we consider the last feature map in convolution-based architectures and the last cls token before the classification head in ViT-Tiny as the penultimate layer, respectively. In Fig. 8 we provide a visual overview of the different mode stitching settings and their relationship to FuLA.

# E AGREEMENT PROBABILITY

Given a front model $f : \mathcal{X} \to \mathcal{W}$ and an end model $g : \mathcal{X} \to \mathcal{W}$, we compute the agreement probability as:

$$\text{agreement probability} = \frac{f(x)^T g(x)}{\max(f(x)^T f(x), (g(x)^T g(x))}. \tag{8}$$

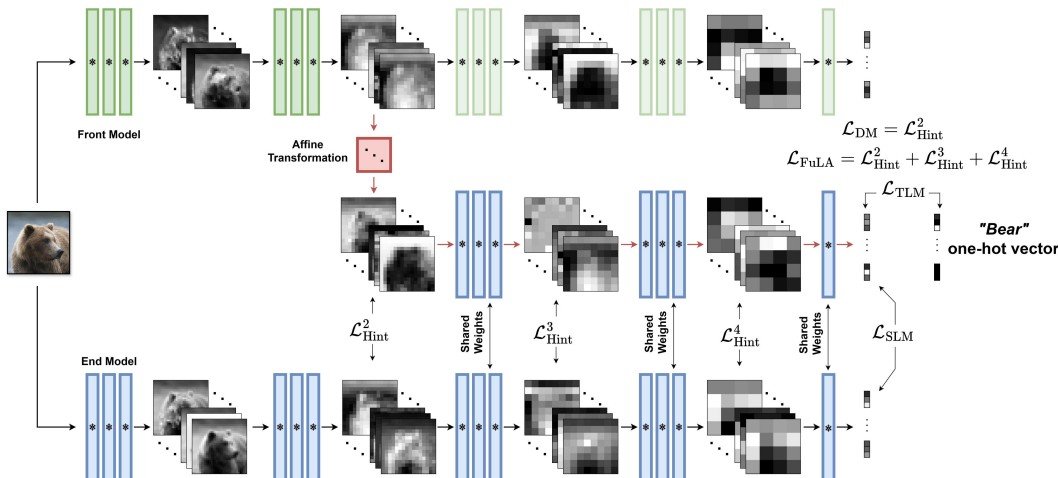

Figure 8: Overview of the different objectives in model stitching for functional similarity evaluation.

The normalization term (i.e the denominator) was introduced to avoid confidently predicted samples dominating the agreement probability.

## F IRIs SANITY TEST

To ensure that the $\mathcal{D}_{\text{train}}^{\text{IRIs}}$ resembles the original data in $\mathcal{D}_{\text{train}}$ with respect to the front model, at the stitch level, we perform model stitching between identical models (i.e., identical stitching) where the front model was given the original input in $\mathcal{D}_{\text{train}}$ and the end model was given the corresponding sample from the $\mathcal{D}_{\text{train}}^{\text{IRIs}}$ and vice versa. Given that the front and end models are identical, any deviation from the optimal agreement can be attributed to the data being perceived differently by the same model. In Fig. 9 we observe that in all cases the stitching was almost perfect for the DM objectives indicating that the $\mathcal{D}_{\text{train}}^{\text{IRIs}}$ and $\mathcal{D}_{\text{train}}$ are almost identical at the stitch-level allowing us to probe for backward compatibility under Req. B. For the most part. high agreement was also achieved for the other settings as well despite the IRIs data not being explicitly optimized for invariance at the layers following the stitching.

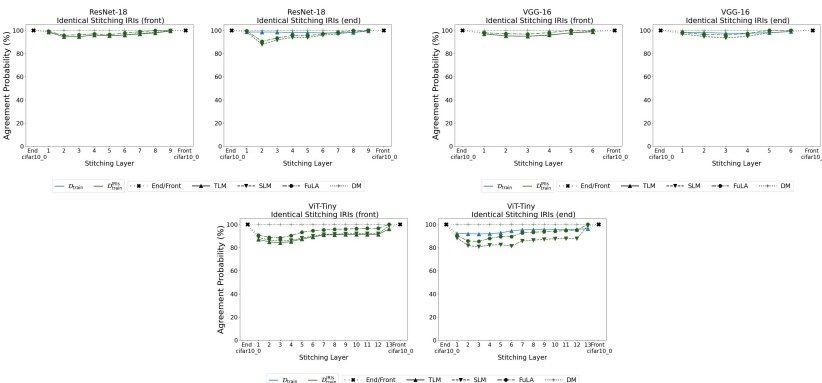

Figure 9: Stitching between identical models using corresponding pairs from $\mathcal{D}_{\text{train}}^{\text{IRIs}}$ and $\mathcal{D}_{\text{train}}$ (CIFAR-10).

## G RESULTS ON ADDITIONAL ARCHITECTURES

In this section we repeat the experiments conducted in the paper across all architectures (i.e., ResNet-18, VGG-16, ViT-Tiny and Vit-Base) and dataset considered.

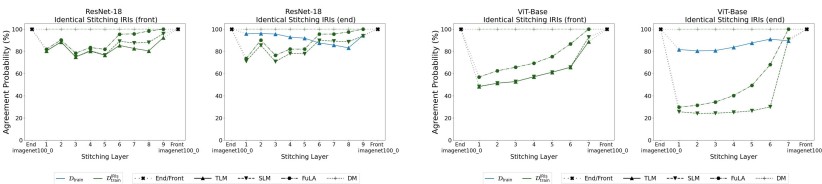

Figure 10: Stitching between identical models using corresponding pairs from $\mathcal{D}_{\text{train}}^{\text{IRIs}}$ and $\mathcal{D}_{\text{train}}$ (ImageNet-100).

## G.1 FUNCTIONAL SIMILARITY ACROSS INFORMATION VARIANTS

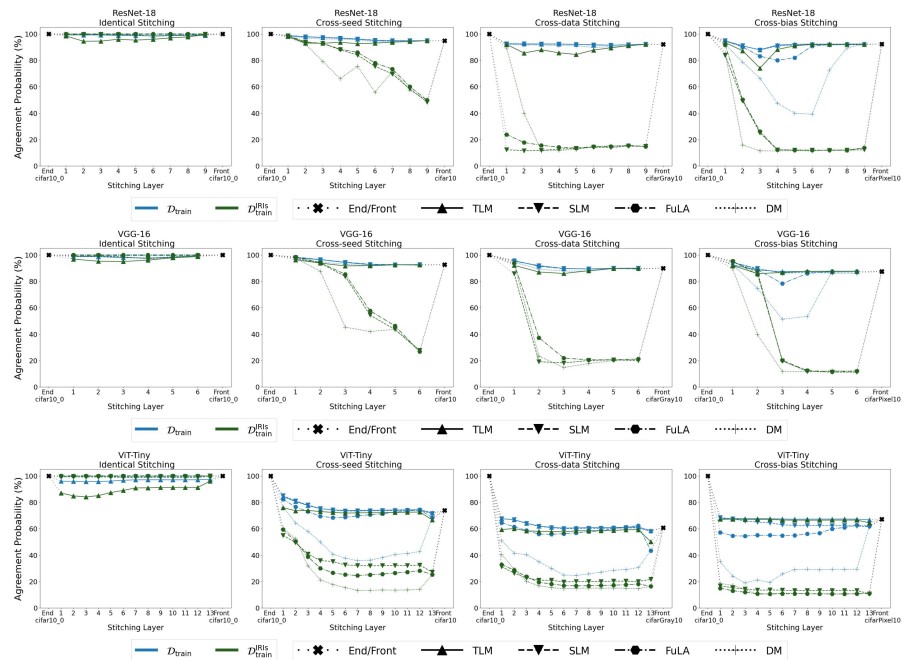

Figure 11: Stitching between models relying on different information (CIFAR-10).

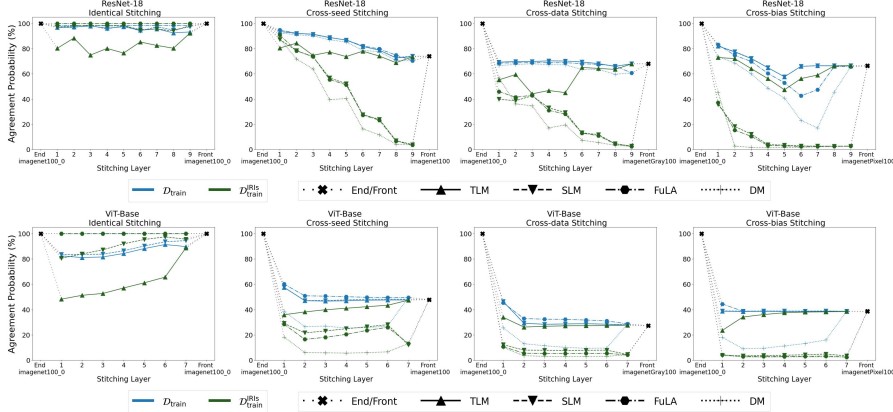

Figure 12: Stitching between models relying on different information (ImageNet-100).

In Figs. 11 and 12 we observe similar trends across all architectures and datasets, where in regular model stitching the agreement probability is roughly interpolated between the end and front models.

Although DM constitutes a frequent exception to this trend, high similarity is recovered by the deeper layers for ResNet-18 and VGG-16 or by the last layer for ViT-Tiny and ViT-Base. On the other hand, under the invariance-aware model stitching, the drops in agreement are sharper in cases where we would intuitively expect lower similarity. Among the invariance-aware objectives, I-FuLA and I-SLM achieve higher similarity in convolution-based architectures and transformer-based respectively. However, we note that in the Identity stitching configuration in ViT-Base, we observe that I-SLM fails to retrieve the optimal transformation in which case it is trivial. We observe the effect both for ResNet-18 and Vit-Base when trained on ImageNet-100, however the failure effect is significantly pronounced for Vit-Base in which case we consider the sanity check to have failed (see Fig. 12, second row, first column).

## G.2 FUNCTIONAL SIMILARITY UNDER UNSEEN PREDICTIVE INFORMATION

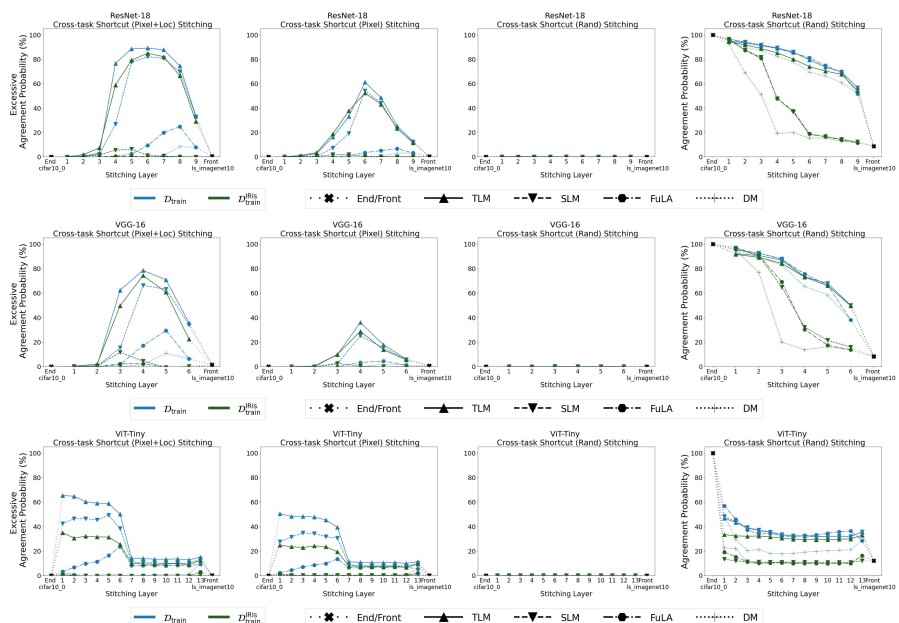

Figure 13: Cross-task stitching under shortcuts of varying availability (CIFAR-10)

Figs. 13 and Figs. 14 suggests similar trends across all architectures, where under regular model stitching all settings are susceptible to relying on shortcuts for achieving alignment. In contrast, invariance-aware model stitching is resistant to learning these shortcuts, with only I-SLM (in convolution-based architectures) engaging in shortcut learning, and even then to a considerably lesser extent compared to SLM. Note that TLM, even when trained on the IRIs data still behaves similar to the regular methods as it does not incorporate perception signal from the end model and therefore can meaningfully turn into an invariance-aware setting.

The evidence from the cross-task stitching configuration highlights an undesirable property of model stitching for functional similarity evaluation. Specifically, the stitching layer should only align representations and should not engage in learning, as argued by Bansal et al. (2021). Identifying even a single configuration in which this undesired behavior emerges is sufficient to demonstrate that it can occur and, consequently, undermine the reliability of similarity evaluation. Nevertheless, we repeated the same analysis using models trained on the same task (i.e., cross-seed stitching) and found that the stitching layer still engaged in learning but the effect is significantly less pronounced both in terms of consistency and intensity as shown in Fig. 15.

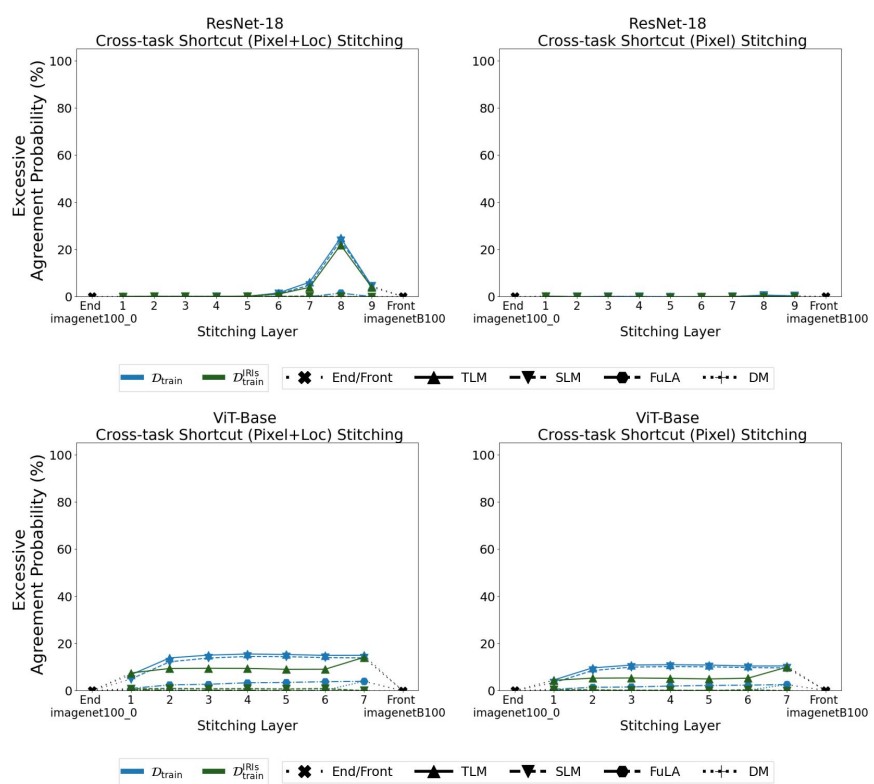

Figure 14: Cross-task stitching under shortcuts of varying availability (ImageNet-100).

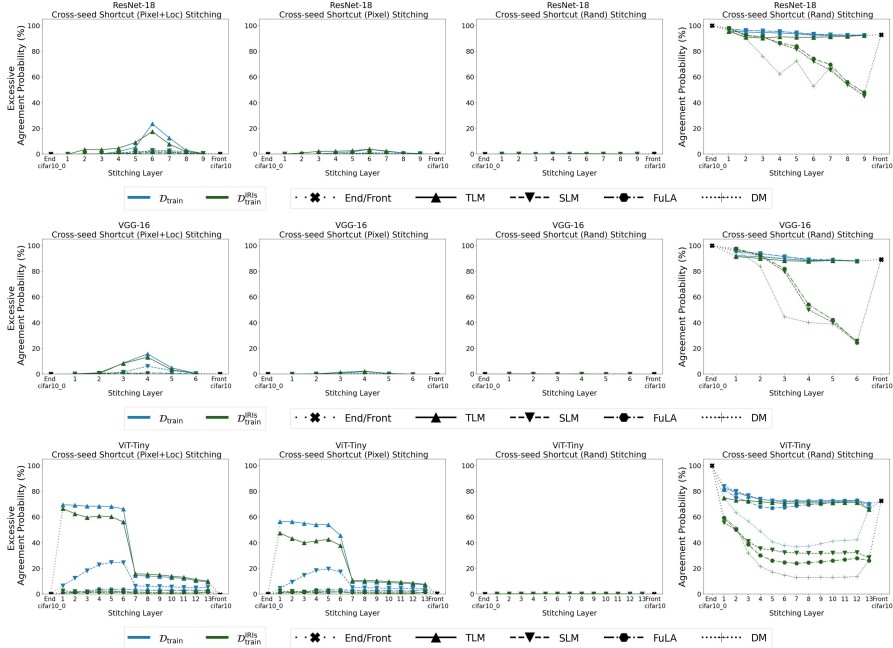

Figure 15: Cross-seed stitching under shortcuts of varying availability (CIFAR-10).

## G.3 Functional Similarity between Robust and non-Robust models

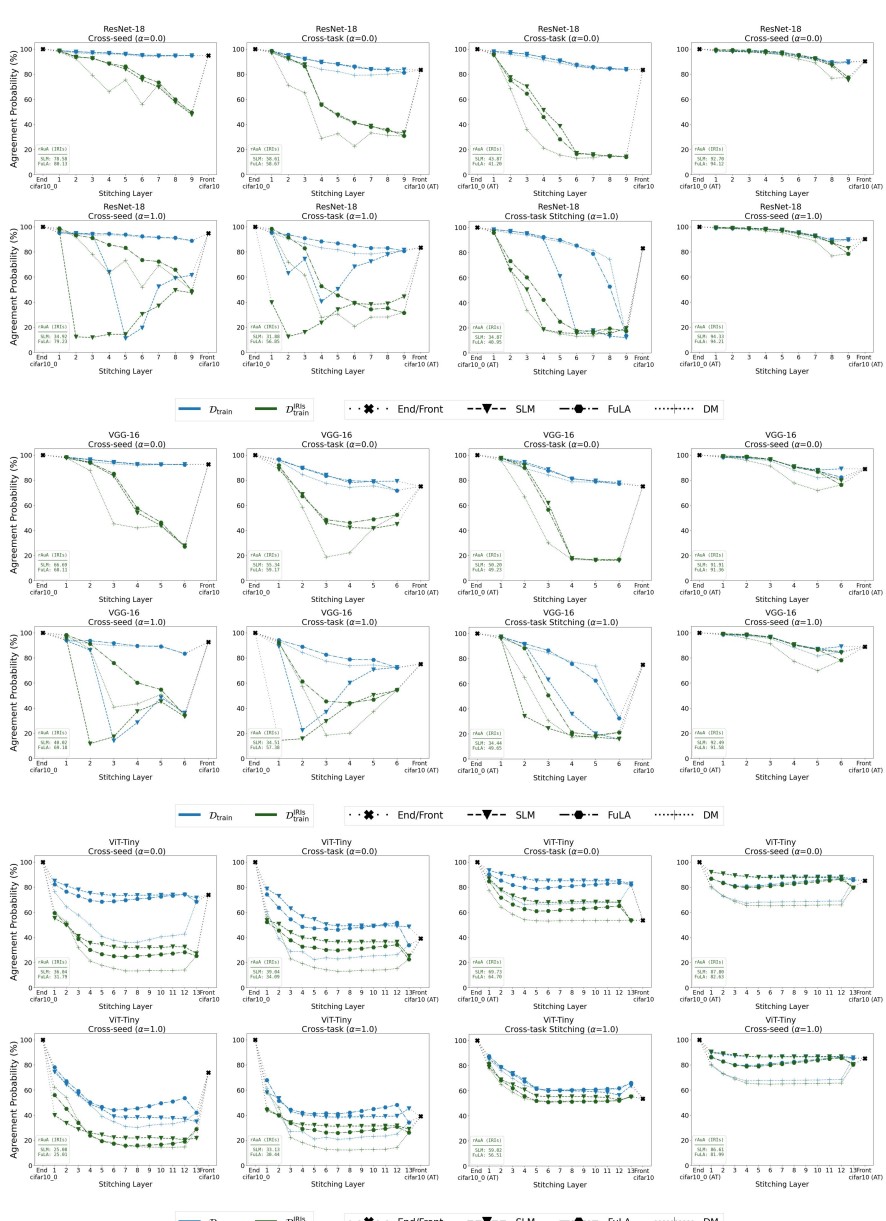

Figure 16: Stitching between robust and non-robust models (CIFAR-10). The Area under Agreement (AuA) measurements refer to the average agreement probability achieved between the stitching layers (excluding the end and front points).

In Fig. 16, we provide results on robust-to-non-robust model stitching. The analysis carried out on ResNet-18 applies to VGG-16 as well, where we conclude that I-FuLA consistently achieves higher similarity than I-DM while remaining stable across all configurations. In contrast, I-SLM displays erratic behavior in configurations where there is a mismatch between the end model and the objective used to establish alignment.

For ViT-Tiny, we again observe that invariance-aware stitching achieves a lower degree of similarity. Additionally, we note that regular model stitching fails to identify that robust to robust configurations are more similar than non-robust to robust configurations for both clean and adversarial settings. This property, however, is captured under invariance-aware stitching and was also observed by Nanda et al.

(2022) from the representation-similarity standpoint. In contrast to convolution-based architectures, I-SLM behaves stably while generally achieving higher similarity compared to I-FuLA, rendering the former the overall better setting for ViT-Tiny.

Repeating the same analysis for ImageNet-100 or ResNet-18 and Vit-Base found in Fig. 17 leads to similar conclusion.

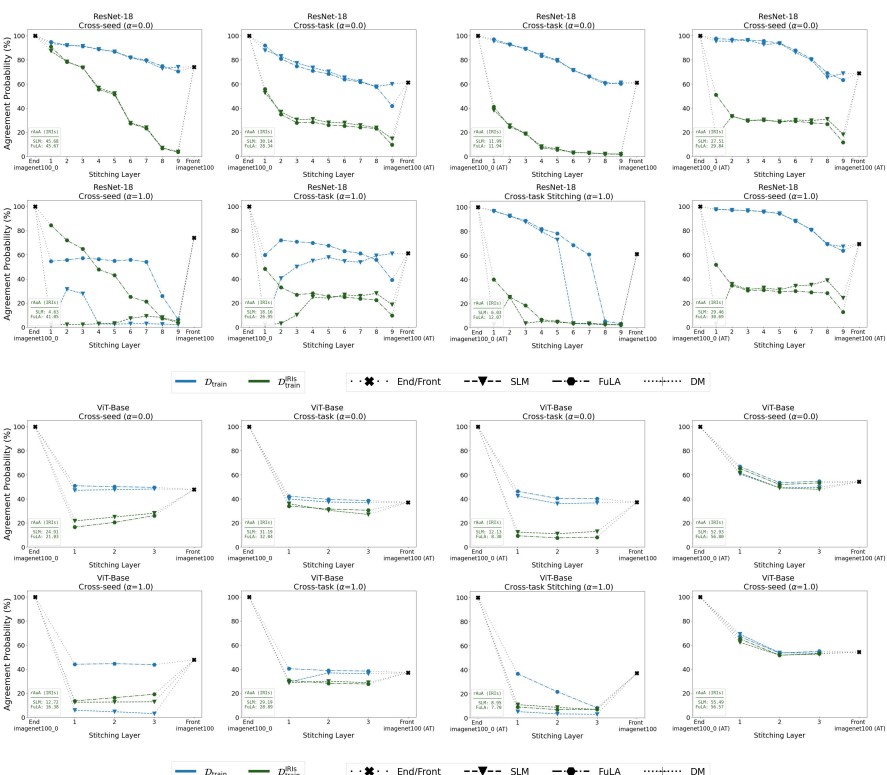

Figure 17: Stitching between robust and non-robust models (ImageNet-100). The Area under Agreement (AuA) measurements refer to the average agreement probability achieved between the stitching layers (excluding the end and front points).

## H RELATIONSHIP BETWEEN FULA AND CKA

In this section, we compared (I)-FuLA and (I)-CKA where we were mostly interested to identify which insight are unique to I-FuLA compared to CKA and I-CKA. For fair comparison we scaled (I)-FuLA such that $0\%$ corresponds to random chance performance (i.e, lowest possible agreement probability). Overall the results verify that both CKA and I-CKA are difficult to interpret as their values do not bear any grounded meaning. Overall, we observe that I-FuLA and I-CKA can deviate significantly with the former displaying more erratic behavior.

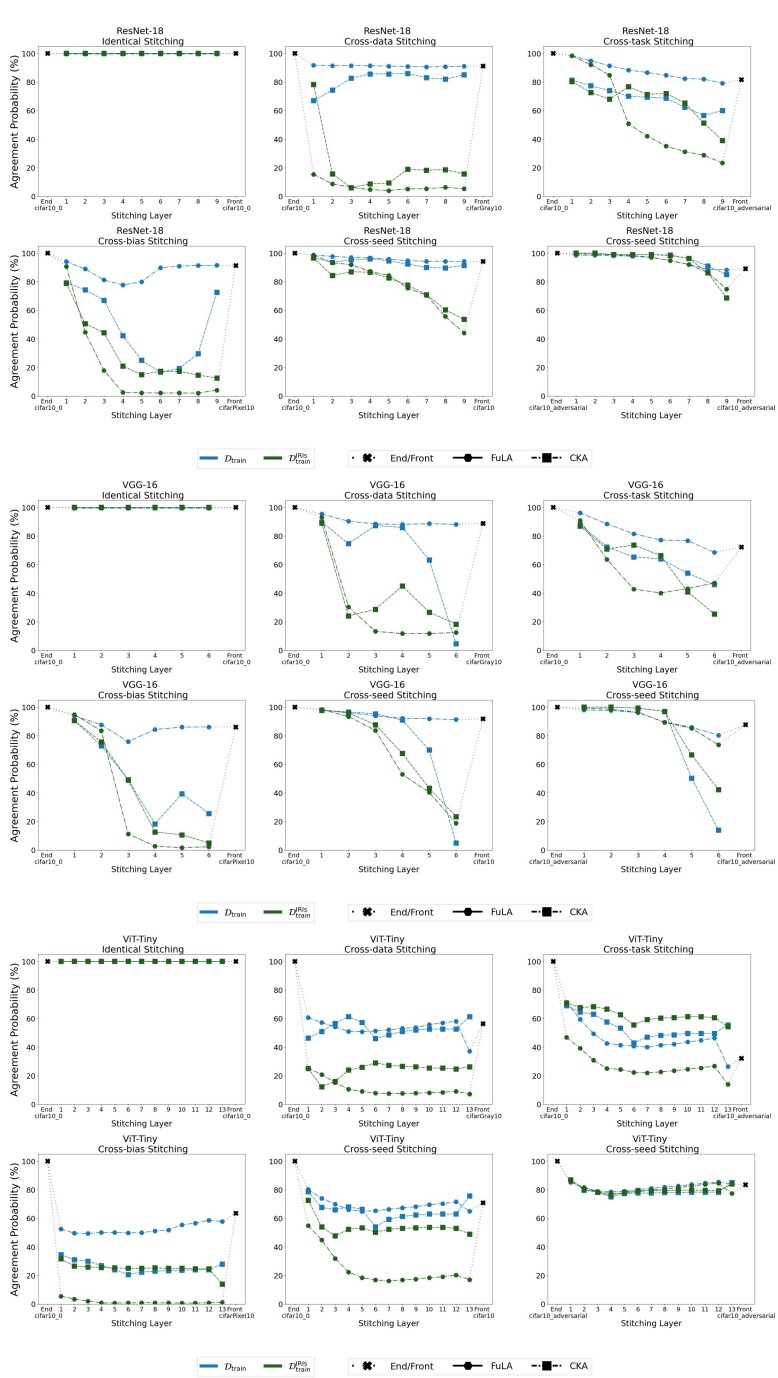

Figure 18: Comparing FuLA and CKA (CIFAR-10).

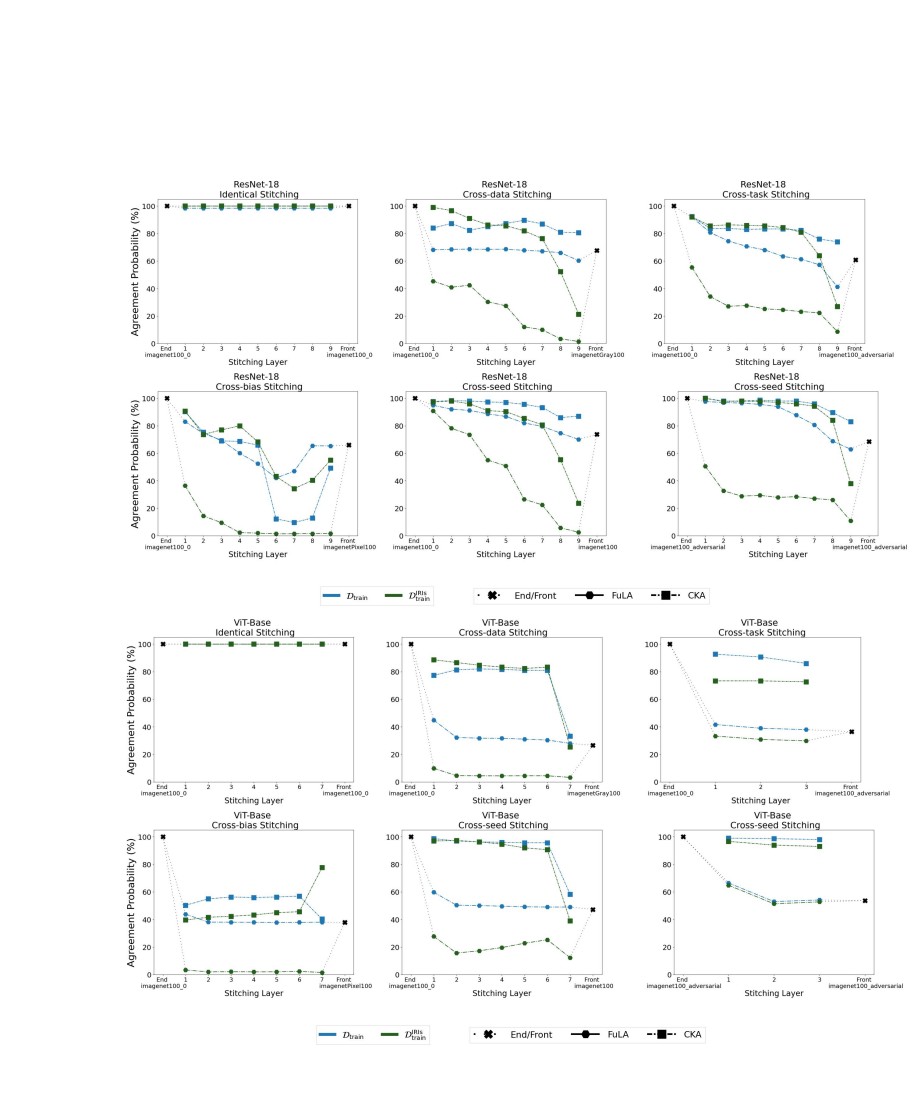