# OpenReview forum: "Model Stitching by Invariance-aware Functional Latent Alignment"
_ICLR.cc/2026/Conference — Submitted to ICLR 2026_

### Official Review · Reviewer_ZFEj · 2025-10-26

**Soundness:** 3
**Presentation:** 2
**Contribution:** 3
**Rating:** 6
**Confidence:** 5

**Summary:**

The paper considers a few recently discovered issues with representation similarity characterizations, that indicate that they are either overly permissive (like stitching with task loss matching, where the stitching layer can get "too creative" in achieving a good task loss) and too penalizing (eg stitching with direct matching, where there is too much focus on fitting a single layer as closely as possible, without regard of the dynamics of the network as a whole).

The paper proposes two techniques. One is a new loss for stitching called functional latent alignment (fula) that involves matching all the layers after the stitching layer, and the other is the utilisation of identically represented inputs as a means to examine whether the inputs that are represented similarly indeed are processed similarly.

The paper then presents an empirical evaluation of the method and compares it with known baselines, showing that the indentically represented inputs are indeed a good tool for separating real difference from "cheating" stitchings.

**Strengths:**

The paper introduces very interesting ideas about fixing the current stitching approaches in order to get a cleaner insight into representation similarity. The main idea seems to be that one needs to look at both the representations in each layer as well as the expected invariances while propagating the representations.

The empirical evaluation is interesting and indeed supports the claims that the proposed techniques add a new, useful perspective.

**Weaknesses:**

The main problem is with the presentation. The paper is very hard to read, even for someone who is familiar with the area. The paper is extremely dense; there is a lot of content packed in a limited space, and so things are not sufficiently motivated, explained, and discussed. The plots are extremely small, and even in color, they are difficult to read. It requires a lot of concentration to understand what the plots show and how the experiments were conducted. At the same time, Figures 1 and 2 take up a lot of space, while I found them a lot more confusing than helpful. Even after understanding the text, I still had a hard time making sense of these plots. I personally don't think they are necessary, at least in the main text, and then you could have larger plots and more words to explain what is going on.

As for the method, it is evident that req B does the heavy lifting, while FULA is not that different from SLM. So I was not entirely convinced that FULA is even necessary. I think the idea of input invariance is the key here. However, I found the motivation of req B less clear, I think it would need some more support and explanation. Even sec 2.3.1 is quite confusing because it is not clear how we take care of req B exactly. (Later becomes somewhat clearer, but essentially just from the way you construct the plots.)

**Questions:**

Are there any cases when FULA is clearly necessary and "better" (in the sense of some sanity checks) than SLM?

You promise at some point that some sanity checks are being used (lacking any formal "oracle), which is fine, but then you do not seem to state your sanity checks clearly and up front. What are these?

---

> ### Author Response · Authors · 2025-12-03
> **Responding to ZFEj (1/2)**
>
> Dear ZFEj, thank you for taking the time to review our work and provide valuable feedback. Please find our responses to the points you raised below.
>
> **Weaknesses**
>
> >  W.1 The paper is extremely dense; there is a lot of content packed in a limited space, and so things are not sufficiently motivated, explained, and discussed.
>
> Based on the feedback received, we have made updates towards improving the clarity of our arguments and the presentation of our results.
>
> > W.1 (cont.) The plots are extremely small, and even in color, they are difficult to read.
>
> We have improved the design of stitching plots by increasing the font and removing the legend outside the figures.
>
> > W.1 (cont.) At the same time, Figures 1 and 2 take up a lot of space, while I found them a lot more confusing than helpful. Even after understanding the text, I still had a hard time making sense of these plots. I personally don't think they are necessary, at least in the main text, and then you could have larger plots and more words to explain what is going on.
>
> We find Fig.1 (now renamed to Fig.2) to be important to support our conceptual arguments in Sec.1 and therefore we keep it in the main text. As for Fig.2 (now renamed to Fig.8), we have moved it to supplementary material and used the extra space to improve the overall clarity of our work.
>
> &nbsp;
>
> > W.2  As for the method, it is evident that req B does the heavy lifting, while FULA is not that different from SLM. So I was not entirely convinced that FULA is even necessary. I think the idea of input invariance is the key here.
>
> Indeed proposing to probe for backward compatibility (Req. B) is our contribution with the biggest impact. However, there are cases, where latent-level forward compatibility (Req. A) is also relevant, kindly refer to our response to your question below.
>
> > W.2 (cont.) However, I found the motivation of req B less clear, I think it would need some more support and explanation.
>
> Towards improving the motivation of backward compatibility (i.e., Req. B), in Sec.1, we introduced a toy functional alignment example (see Fig.1).
>
> > W.2 (cont.) Even sec 2.3.1 is quite confusing because it is not clear how we take care of req B exactly. (Later becomes somewhat clearer, but essentially just from the way you construct the plots.)
>
> In Sec.2.3.1, we explain both the construction of the IRIs dataset ($\mathcal{D}^\text{IRIs}_\text{train}$) and how Req. B is probed in practice. Based on this, we are not sure which aspect of our procedure for probing Req. B was unclear.

---

> ### Author Response · Authors · 2025-12-03
> **Responding to ZFEj (2/2)**
>
> **Questions**
>
> > Q.1 Are there any cases when FULA is clearly necessary and "better" (in the sense of some sanity checks) than SLM?
>
> During the rebuttal, we expanded the experimental scope and found that I-SLM fails the identity-stitching sanity check for the ViT-Base architecture. Our findings can be summarized as follows:
>
> - Cases where I-FuLA fails a sanity check: None
>
> - Cases where I-SLM fails a sanity check: ViT-Base (identity stitching)
>
> - Among architectures passing all sanity checks, I-FuLA performs best on: ResNet-18, VGG-16, ViT-Base
>
> - Among architectures passing all sanity checks, I-SLM performs best on: ViT-Tiny
>
>
> Thus, in terms of sanity checks, I-FuLA is clearly better than I-SLM for ViT-Base.
> Performance-wise, when both methods pass the sanity checks, I-SLM and I-FuLA are comparable in most cases. However, I-FuLA is substantially better when measuring functional similarity on adversarial samples for non-robust CNN end models (e.g., see Fig.6, second row, first and second columns).
>
> &nbsp;
>
> > Q.2 You promise at some point that some sanity checks are being used (lacking any formal "oracle"), which is fine, but then you do not seem to state your sanity checks clearly and up front. What are these?
>
> We refer to the experiments in Secs.3.1 [Functional similarity across information variants] and 3.2 [Functional similarity under unseen predictive information] as sanity checks, as we have an intuitive notion of how a meaningful functional similarity metric should behave but lack any formal ground truth (with the identity stitching being the only exception to that). For example, in Sec.3.1, we intuitively expect two independently trained models on regular data to be more functionally similar than a model trained on regular data and one trained on shortcuts.
>
> To avoid confusion, we introduce explicit reference to Secs.3.1 and 3.2 when referring to these sanity checks for the first time in the paper.

---

### Official Review · Reviewer_HTsy · 2025-10-27

**Soundness:** 2
**Presentation:** 1
**Contribution:** 2
**Rating:** 4
**Confidence:** 3

**Summary:**

This paper investigates the problem of functional similarity in deep neural networks using the model stitching paradigm. The authors argue that existing stitching methods, which primarily focus on "forward compatibility" (i.e., maintaining task performance), can be misleading. They show that these methods often find high similarity even between models trained on different "information cues" (e.g., color vs. texture). To address this, the paper introduces two requirements for a more meaningful similarity measure: (1) latent-level forward compatibility, ensuring internal representations transition similarly after stitching, and (2) "backward compatibility", ensuring inputs that are invariant to the first model are also treated similarly by the second.

The paper proposes a new stitching objective, Functional Latent Alignment (FuLA), to enforce forward compatibility, and an "invariance-aware" training setup using Identically Represented Inputs (IRIs) to probe for backward compatibility.

**Strengths:**

- The paper addresses the critical problem of understanding and quantifying the similarity between neural network representations, which is fundamental to interpretability and model understanding.

- The paper's primary conceptual contribution is the introduction of "backward compatibility" as a necessary condition for meaningful similarity.

- The authors conduct a comprehensive set of experiments across multiple architectures (ResNet, VGG, ViT) and under various conditions, including different data cues and model robustness settings.

**Weaknesses:**

1. The paper is difficult to follow. The writing is often dense, and key concepts like "information cues" are used without a precise definition. Most importantly, Figures 4, 5 and 6 are not well-explained in the caption, are hard to interpret and it is extremely hard to match the trace to the correct item in the legend.

2. The interpretation of some results is questionable. For example, in the cross-data stitching experiment (CIFAR-RGB vs. CIFAR-grayscale), the sharp performance drop for I-FuLA is presented as a success. However, one could argue that since the underlying image content is the same, with different augmentations, a good similarity metric should yield high similarity. The paper does not defend *why* this sharp drop is a desirable property.


3. The paper's core premise, that models trained on different "information cues" should be considered functionally dissimilar, is not sufficiently motivated. This stance appears to contradict a growing body of work suggesting that models can and should learn compatible or geometrically aligned representations if the underlying data semantics are the same (e.g., the Platonic Hypothesis). The paper fails to adequately position itself against this literature, for example the works on relative representations are cited but sligthly misinterpreted (Moschella et al., 2022, Cannistraci et al., 2023): they have already been used as invariance-aware similarity measure between representations  (e.g., Section 4.1 in Moschella et al) and not only for model stitching.


4. The analysis is missing crucial ablations. The expressivity of the stitching transformation S is fixed to a linear layer. However, the capacity of this transformation is a critical factor that could heavily influence the stitching outcome. An analysis with different capacities (e.g., identity, a small MLP) is needed to disentangle the effects of the stitching objective from the effects of the transformation's capacity.

**Questions:**

- Could the authors clarify their position with respect to the emerging similarity literature? These works suggest that compatible representations should emerge from data with shared semantics, even if the inputs differ (e.g., images and captions). Why should models trained on grayscale vs. RGB images be considered functionally dissimilar?

- Regarding the cross-data stitching experiment (Fig. 4, "Cross-data"), the authors present the sharp decline in performance for I-FuLA as a positive outcome. Further elaboration is needed on why this is a desirable result. An alternative viewpoint is that the models should be able to find common ground, as the semantic content is largely preserved between colored and grayscale images.

- How do the results and conclusions change when varying the expressivity of the stitching transformation S (e.g., using a multi-layer perceptron instead of a single linear layer)? It seems possible that a more expressive stitch layer could overcome the dissimilarities that I-FuLA is designed to detect, which would challenge the paper's conclusions.

- The paper would benefit from a more high-level, intuitive explanation of the core concepts before diving into the formal notation. This would make the work more accessible.

---

> ### Author Response · Authors · 2025-12-04
> **Responding to HTsy 1/3**
>
> Dear HTsy, thank you for your time and your thorough feedback. Please find our responses to the points you raised below.
>
> **Weaknesses**
>
> > W.1 Figures 4, 5 and 6 are not well-explained in the caption, are hard to interpret and it is extremely hard to match the trace to the correct item in the legend.
>
> We have updated the stitching plots by incorporating feedback from all reviewers. Namely, we removed the legends outside the figures as well as increased the font size. Additionally we made the captions in Figs.4,5 and 6 more verbose for clearer result presentation.
>
> &nbsp;
>
> > W.2 The interpretation of some results is questionable. For example, in the cross-data stitching experiment (CIFAR-RGB vs. CIFAR-grayscale), the sharp performance drop for I-FuLA is presented as a success. However, one could argue that since the underlying image content is the same, with different augmentations, a good similarity metric should yield high similarity. The paper does not defend why this sharp drop is a desirable property.
>
> We can not make claims about the cross-data stitching configuration (RGB vs. grayscale) in isolation. Our argument is grounded in the cross-bias stitching experiment (Fig. 4, panel four), where the front model performs near random chance (see footnote 3) when shortcuts are absent. This establishes that a model trained with shortcuts relies on fundamentally different cues than one trained on regular data. Consequently, high functional similarity between these models would be misleading.
> In particular, the penultimate layers of a model trained on RGB data and one trained on shortcuts can not realize the same functionality, for example, transfer learning from either layer would yield very different performance.
> Based on this understanding, we revisit cross-data stitching, we do not interpret the performance drop as a success per se. Instead, we regard the drop in the cross-bias configuration as a reliable indicator of meaningful functional divergence, and we report the behavior observed in the cross-data configuration.

---

> ### Author Response · Authors · 2025-12-04
> **Responding to HTsy 2/3**
>
> > W.3 The paper's core premise, that models trained on different "information cues" should be considered functionally dissimilar, is not sufficiently motivated.
>
> In our work, we share the same view as [3] and argue that a *meaningful* notion of functional similarity should measure the extent to which two models rely on similar input patterns (i.e., information cues) to solve their respective tasks. As a motivating example, consider a well-trained model that captures task-relevant features and another model that relies on opportunistic shortcuts, these two models do not perform the same internal functionalities, and therefore a meaningful functional similarity measure should be able to tell them apart.
>
> > W.3 (cont.) This stance appears to contradict a growing body of work suggesting that models can and should learn compatible or geometrically aligned representations if the underlying data semantics are the same (e.g., the Platonic Hypothesis).
>
> First we would like to note that in Platonic Hypothesis [9] paper convergent representations were observed with respect to mutual nearest neighbors while it was also reported that the convergence trend was weak under CKA. That is, such convergent trends are not universal can be contradictory among representation similarity metrics, let alone functional similarity metrics that measure similarity with respect to the subspace relevant to the functionality (i.e, the task).
>
> Even under the assumption that models converge to representation of similar geometric structure, they may do so while differing in the functions they learn. Based on that it is possible that convergent representation can co-exist with functional divergence.
>
>
> > W.3 (cont.) The paper fails to adequately position itself against this literature
>
> We have included a paragraph in Sec.3.4 positioning our work in relation to the body of literature suggestive of structurally convergent representation.
>
> > W.3 (cont.) for example the works on relative representations are cited but sligthly misinterpreted (Moschella et al., 2022, Cannistraci et al., 2023): they have already been used as invariance-aware similarity measure between representations (e.g., Section 4.1 in Moschella et al) and not only for model stitching.
>
> We characterize the setting that probes for backward compatibility (i.e., Req. B) as invariance-aware because it penalizes similarity when alignment can not be established between two models through their invariances.
>
> The referenced works suggest that independently trained models can be successfully stitched using the notion of relative representations. The invariance characterization in Moschella et al. [10] refers to the latent representations themselves, that is, the representations learned via relative representations are invariant to “stochastic factors in the training process” [10]. This invariance explains why relative representations are highly similar across different word embeddings (i.e., independently trained models), as discussed in Section 4.1 of Moschella et al. [10].
> These two notions of invariance are distinct and should not be conflated.
> &nbsp;
>
> > W.4 The analysis is missing crucial ablations. The expressivity of the stitching transformation S is fixed to a linear layer. However, the capacity of this transformation is a critical factor that could heavily influence the stitching outcome. An analysis with different capacities (e.g., identity, a small MLP) is needed to disentangle the effects of the stitching objective from the effects of the transformation's capacity.
>
> Conceptually identical point was also raised by Review MHaa which we responded to as follows:
>
> Modelling the stitching layer as an affine transformation is standard practice in the functional similarity evaluation literature [1,3,4,5,6].
>
>  In our work, we follow this standard, ensuring that our contributions and insights remain relevant to the field.
> The motivation is that the transformation should be flexible enough to capture non-trivial mappings without increasing the capacity of the stitched composition. For example, an insufficiently flexible transformation would fail to recover a simple permutation mapping, under-reporting trivial functional alignment. Conversely, an overly flexible stitching transformation could memorize associations and report high functional similarity even when none exists. In that sense, an affine transformation is generally viewed as striking the right balance for the stitching layer [5].

---

> ### Author Response · Authors · 2025-12-04
> **Responding to HTsy 3/3**
>
> **Questions**
>
> > Q.1 Could the authors clarify their position with respect to the emerging similarity literature? These works suggest that compatible representations should emerge from data with shared semantics, even if the inputs differ (e.g., images and captions). Why should models trained on grayscale vs. RGB images be considered
>
> Kindly refer to our response under W.3.
>
> &nbsp;
>
> > Q.2 Regarding the cross-data stitching experiment (Fig. 4, "Cross-data"), the authors present the sharp decline in performance for I-FuLA as a positive outcome. Further elaboration is needed on why this is a desirable result. An alternative viewpoint is that the models should be able to find common ground, as the semantic content is largely preserved between colored and grayscale images.
>
> Kindly refer to our response under W.2.
>
> &nbsp;
>
> > Q.3 How do the results and conclusions change when varying the expressivity of the stitching transformation S (e.g., using a multi-layer perceptron instead of a single linear layer)? It seems possible that a more expressive stitch layer could overcome the dissimilarities that I-FuLA is designed to detect, which would challenge the paper's conclusions.
>
> Kindly refer to our response under W.4.
>
> &nbsp;
>
> > Q.4 The paper would benefit from a more high-level, intuitive explanation of the core concepts before diving into the formal notation. This would make the work more accessible.
>
> Towards providing a more accessible and intuitive explanation of the core concepts, in Sec.1, we introduced a toy functional alignment example (see Fig.1). Through this example, we crystallize the notions of forward and backward compatibility by contrasting forward-aligned with forward- and backward-aligned functions.
>
> &nbsp;
>
> **References:**
>
> [1] Bansal, Yamini, Preetum Nakkiran, and Boaz Barak. "Revisiting model stitching to compare neural representations." Advances in neural information processing systems 34 (2021): 225-236.
>
> [3] Smith, Damian, Harvey Mannering, and Antonia Marcu. "Functional Alignment Can Mislead: Examining Model Stitching." Forty-second International Conference on Machine Learning.
>
> [4] Balogh, András, and Márk Jelasity. "How not to stitch representations to measure similarity: Task loss matching versus direct matching." Proceedings of the AAAI Conference on Artificial Intelligence. Vol. 39. No. 15. 2025.
>
> [5] Balogh, András, and Márk Jelasity. "On the functional similarity of robust and non-robust neural representations." International Conference on Machine Learning. PMLR, 2023.
>
> [6] Csiszárik, Adrián, et al. "Similarity and matching of neural network representations." Advances in Neural Information Processing Systems 34 (2021): 5656-5668.
>
> [9] Huh, Minyoung, et al. "The platonic representation hypothesis." arXiv preprint arXiv:2405.07987 (2024).
>
> [10] Moschella, Luca, et al. "Relative representations enable zero-shot latent space communication." arXiv preprint arXiv:2209.15430 (2022).

---

### Official Review · Reviewer_MHaa · 2025-10-28

**Soundness:** 1
**Presentation:** 1
**Contribution:** 2
**Rating:** 2
**Confidence:** 4

**Summary:**

This paper introduces I-FuLA, a new model stitching method to better measure functional similarity. It combines a novel objective, Functional Latent Alignment (FuLA), with an "invariance-aware" setting that learns the alignment on inputs with identical internal representations (IRIs). Experiments show this method provides a more meaningful similarity score, as it can distinguish between models trained on different visual cues and avoids exploiting spurious shortcuts. This leads the authors to conclude that robust and non-robust networks are less functionally similar than previously believed.

**Strengths:**

* **S1**: The paper tackle an interesting problem of stitching different neural networks and evaluating latent similarities.
* **S2**: The paper introduce the backward compatibility, as a new and useful rule for measuring latent similarity.

**Weaknesses:**

* **W1. Clarity and Presentation**: The major weakness is that the paper is dense and can be difficult to follow. It doesn't follow a clear and linear story. Additional, the core concepts of "forward" and "backward" compatibility, while central to the paper, are not introduced with sufficient clarity early on. The notation, though systematic, adds to the cognitive load. The figures, particularly the "stitching plots," are small and contain a lot of information, making them hard to decipher without extensive cross-referencing with the text. A more guided walkthrough of one of the plots in the main text would have been beneficial.

* **W2. Limited Experimental Scope**: The experimental validation is conducted on relatively small-scale datasets (CIFAR-10 and a 10-class subset of ImageNet) and primarily with one architecture (ResNet-18). While VGG-16 and ViT-Tiny are included in the appendix, the main claims are built on the ResNet-18 results. The findings would be much more compelling if demonstrated on larger-scale benchmarks (e.g., the full ImageNet-1k) and with a more diverse set of modern architectures, especially larger Transformers.

* **W3. Novelty and Contribution Statement**: The paper's primary novelty lies in formalizing the "backward compatibility" requirement and using IRIs to test it. However, this could be stated more directly in the introduction and contributions list. The introduction of I-FuLA, while new, appears to be a secondary contribution, as the "invariance-aware" setting is what drives most of the significant results. The paper could be improved by more clearly delineating the impact of each of these two contributions.

* **W4. Comparison to Other Metrics and Related Work**: The paper does not compare its similarity findings to other representation similarity metrics like Centered Kernel Alignment (CKA) [1] or to other more recent works such as [2]. Such a comparison would help contextualize their results and clarify what unique insights the notion of "functional similarity" provides over geometric or statistical similarity of representations. Additional, the authors could consider including in the related work section the following model-stitching works [4,5,6].

* **W5. Reproducibility**: Providing the code would be essential for the community to verify the results and build upon this work.

* **W6. Subjectivity of "Meaningful Similarity**: A core claim of the paper is that it provides a more "meaningful" measure of similarity. However, "meaningful" is never formally defined and is instead based on intuitive sanity checks. While the experiments are convincing, the paper would be stronger if it could connect its measure to a more concrete, objective property, or discuss the philosophical underpinnings of what makes a similarity measure meaningful.


---
[1] Kornblith, Simon, et al. "Similarity of neural network representations revisited." International conference on machine learning. PMlR, 2019.

[2] Fumero, Marco, et al. "Latent functional maps." ICML 2024 Workshop on Geometry-grounded Representation Learning and Generative Modeling. 2024.

[4] Maiorca, Valentino, et al. "Latent space translation via semantic alignment." Advances in Neural Information Processing Systems 36 (2023): 55394-55414.

[5] Cannistraci, Irene, et al. "Bootstrapping parallel anchors for relative representations." ICLR Tiny Paper (2023).

[6] Lähner, Zorah, and Michael Moeller. "On the direct alignment of latent spaces." Proceedings of UniReps: the First Workshop on Unifying Representations in Neural Models. PMLR, 2024.

**Questions:**

* **Q1. Generality for Transformers**: The results indicate that for ViT-Tiny, the proposed I-FuLA is not the optimal setting, and I-SLM is preferred. Does this imply that the definition of "meaningful functional similarity" is architecture-dependent, and that different model families may require different criteria?
* **Q2. Scalability to Larger Datasets**: How would the authors expect these findings to translate to more complex, large-scale benchmarks like the full ImageNet-1k dataset? The generation of the DIRIs dataset seems computationally intensive; is this approach feasible at that scale?
* **Q3. Scalability to Larger Models**: How would the authors expect these findings to translate to larger networks, such as larger Vision Transformers (e.g., ViT-Small/Base/Large) or models like DINO?
* **Q4. Framing of Similarity as a Limitation**: In the abstract, the fact that models trained on different information cues can produce compatible representations is framed as a "critical limitation." Could the authors elaborate on why this is a limitation, rather than an interesting property of neural networks (e.g., demonstrating that different paths can lead to functionally similar solutions)?
* **Q5. On the Role of the Stitching Layer's Capacity**: The experiments use a 1x1 convolutional layer for stitching. Could the results be sensitive to the capacity of this transformation layer? Is it possible that models appear dissimilar simply because a simple affine transformation is insufficient to align their representations, and a more powerful non-linear "stitching function" might reveal deeper similarities?

---

> ### Author Response · Authors · 2025-12-03
> **Responding to  MHaa 1/3**
>
> Dear MHaa, thank you for your thorough review and for the constructive feedback. Please find our responses to the points you raised below.
>
> &nbsp;
>
> **Weaknesses**
>
> > W.1  Non-linear storyline.
>
> We acknowledge that the storyline may appear non-linear. However, we had to partially deviate from a perfectly linear storyline because we rely on multiple sanity tests (a common practice in the related literature [2,3,4]) to build our arguments.
>
> Nevertheless, we found your summary to linearly outline our work, which we take as indication that the storyline is sufficiently coherent. We remain open to suggestions that could help us further linearize it.
>
> > W.1 (cont.) Core concepts.
>
> Towards improving the clarity of the core concepts, in Sec.1, we introduced a toy functional alignment example (see Fig.1). Through this example, we crystallize the notions of forward and backward compatibility by contrasting forward-aligned with forward- and backward-aligned functions.
>
> > W.1 (cont.) Notation.
>
> We closely follow the notation commonly used in the model stitching for functional similarity literature ([4,5,6]), as it ensures the rigor needed to clearly communicate our method.
>
> > W.1 (cont.) Stitching plots.
>
> We have improved the design of stitching plots by increasing the font and removing the legend outside the figures. Additionally, we have extended the example on how to interpret the stitching plot in Sec.3.
>
> &nbsp;
>
> > W.2  Limited experimental scope.
>
> We have expanded upon the scope by performing our analysis on larger dataset and models where the main findings remain intact. (see our response under Questions).
>
> &nbsp;
>
> > W.3 Directly connecting IRIs and backward compatibility.
>
>  We have revised the text to explicitly connect IRIs with backward compatibility in Sec.1 and in the contributions list.
>
> > W.3 (cont.) FuLA secondary contribution.
>
> We acknowledge that the Req. B (i.e., the invariance-aware setting) has a bigger impact compared to Req. A (i.e., FuLA) however we argue that does not invalidate the relevance of our contribution.
>
> In the pre-rebuttal state, we had established that I-FuLA and I-SLM were optimal in CNNs and transformer-based architectures respectively. Post-rebuttal, we expanded upon the experimental scope and found that I-SLM fails at the Identity stitching sanity check, overall our finding can be summarized as:
>
> - Cases where I-FuLA fails a sanity check: None
>
> - Cases where I-SLM fails a sanity check: ViT-Base (identity stitching)
>
> - Among architectures passing all sanity checks, I-FuLA performs best on: ResNet-18, VGG-16, ViT-Base
>
> - Among architectures passing all sanity checks, I-SLM performs best on: ViT-Tiny
>
> Based on these, we argue that our contribution related to Req. A is relevant in both convolution- and transformed-based architectures.
>
> &nbsp;
>
> > W.4 Comparison to representation similarity metrics.
>
>  In our work, we focus on model stitching for functional similarity evaluation and therefore comparison with representation similarity metrics has been out of scope.
>
> However, given your suggestion and the popularity of CKA [2], we compared FuLA and CKA for various configurations under both regular and identical represented inputs (IRIs) data (i.e., FuLA, I-FuLA, CKA[2], I-CKA[7]). Based on these comparisons we verify that unique insights were drawn by (I-)FuLA compared to (I-)CKA. We kindly refer to the Sec.3.4 and Sec.H.
>
> > W.4 (cont.) Recommended related work
>
> We found all three recommended papers to be related to our work and therefore included them in Sec.4.
>
> &nbsp;
>
> > W5. Reproducibility.
>
> We will release our codebase along with detailed instructions on how to reproduce our results upon paper acceptance.
>
> &nbsp;
>
> > W6. Notion of meaningful.
>
> In our work, we share the same view with [3] and argue that a *meaningful* notion of functional similarity should measure the extent to which two models rely on similar input patterns to solve their respective tasks. We have now included this clarification in Sec 1.
>
> > W6. (cont.) Sanity checks.
>
> In the Sec.5, we have acknowledged the lack of *oracle* notion of functional similarity, which necessitated reliance on intuitive sanity checks. However this limitation is inherited to the field of representation similarity measurement.

---

> > ### Author Response · Authors · 2025-12-04
> > **Responding to MHaa 2/3**
> >
> > **Questions**
> >
> > > Q.1 The results indicate that for ViT-Tiny, the proposed I-FuLA is not the optimal setting, and I-SLM is preferred. Does this imply that the definition of "meaningful functional similarity" is architecture-dependent, and that different model families may require different criteria?
> >
> > Given the expanded experimental scope discussed under W.3, I-FuLA passes all sanity checks, whereas I-SLM underperforms only for the ViT-Tiny architecture.
> >
> >
> > Overall, **incorporating backward compatibility (Req. B) in model stitching for functional similarity evaluation appears to be architecture-independent**, as its relevance was consistent across all architectures and datasets considered. On the other hand, **the optimal forward compatibility setting may be architecture-dependent**, however in practice, I-FuLA, albeit not always optimal, passes all sanity checks, making it a more robust choice overall than I-SLM.
> >
> > &nbsp;
> >
> > > Q.2 The generation of the DIRIs dataset seems computationally intensive; is this approach feasible at that scale? How would the authors expect these findings to translate to more complex, large-scale benchmarks like the full ImageNet-1k dataset?
> >
> > Indeed, generating the IRIs dataset is computationally intensive as it scales with the dimensionality and the number of training samples but ultimately feasible given sufficient compute.
> >
> >  To keep the computational load manageable while still evaluating a higher-resolution dataset with more classes, we expanded our analysis to ImageNet-100 [8] on ResNet-18 and ViT-Base.
> >
> > Overall, we found that our conclusions generalized to the larger dataset. Across all architectures and datasets we evaluated, we consistently observed similar behavior patterns. Based on this, we expect that the conclusions and insights of our study generalize to larger models and different training paradigms (e.g., supervised vs. semi-supervised).
> >
> > &nbsp;
> >
> > > Q.3 Scalability to Larger Models: How would the authors expect these findings to translate to larger networks, such as larger Vision Transformers (e.g., ViT-Small/Base/Large) or models like DINO?
> >
> > We extended our analysis to ViT-Base (with high-resolution datasets), where the conclusions were similar to those for ViT-Tiny (with low-resolution datasets). The only meaningful deviation was that, for ViT-Base, I-SLM failed the identity-stitching sanity check, increasing the supporting our proposed method (i.e., I-FuLA).
> >
> > Similarly to earlier, we expect that the conclusions and insights of our study generalize to larger models and/or different training paradigms.
> >
> > &nbsp;
> >
> > > Q.4 Framing of Similarity as a Limitation: In the abstract, the fact that models trained on different information cues can produce compatible representations is framed as a "critical limitation." Could the authors elaborate on why this is a limitation, rather than an interesting property of neural networks (e.g., demonstrating that different paths can lead to functionally similar solutions)?
> >
> > The framing follows Damian et al. [3], where it is argued that functional similarity metrics should distinguish between models that rely on different input patterns to solve their respective tasks.
> >
> > **From a functional similarity perspective:** A well-trained model that captures task-relevant features and a model that relies on opportunistic shortcuts do not perform the same internal functionalities. A meaningful functional similarity metric should be able to differentiate between them.
> >
> > **From a representation similarity perspective:** Both models converging, for example to a homomorphic representation space, can indeed be an interesting property of neural networks, suggesting convergence in the representation sense.
> >
> > Importantly, **these perspectives can co-exist**, that is, neural networks may converge in their representations while differing in the functions they learn. In our work, we focus on the functional perspective, however we argue that both the functional and representation perspectives are relevant for interpreting neural networks and are worth studying.

---

> ### Author Response · Authors · 2025-12-04
> **Responding to MHaa 3/3**
>
> > Q.5 On the Role of the Stitching Layer's Capacity: The experiments use a 1x1 convolutional layer for stitching. Could the results be sensitive to the capacity of this transformation layer? Is it possible that models appear dissimilar simply because a simple affine transformation is insufficient to align their representations, and a more powerful non-linear "stitching function" might reveal deeper similarities?
>
>
> Modelling the stitching layer as an affine transformation is standard practice in the functional similarity evaluation literature [1,3,4,5,6]. In our work, we follow this standard, ensuring that our contributions and insights remain relevant to the field.
>
>
> The motivation is that the transformation should be flexible enough to capture non-trivial mappings without increasing the capacity of the stitched composition. For example, an insufficiently flexible transformation would fail to recover a simple permutation mapping, under-reporting trivial functional alignment. Conversely, an overly flexible stitching transformation could memorize associations and report high functional similarity even when none exists. The motivation is that the transformation should be flexible enough to capture non-trivial mappings without increasing the capacity of the stitched composition. For example, an insufficiently flexible transformation would fail to recover a simple permutation mapping, under-reporting trivial functional alignment. Conversely, an overly flexible stitching transformation could memorize associations and report high functional similarity even when none exists. In that sense, an affine transformation is generally viewed as striking the right balance for the stitching layer [5].
>
> &nbsp;
>
> **References:**
>
> [1] Bansal, Yamini, Preetum Nakkiran, and Boaz Barak. "Revisiting model stitching to compare neural representations." Advances in neural information processing systems 34 (2021): 225-236.
>
> [2] Kornblith, Simon, et al. "Similarity of neural network representations revisited." International conference on machine learning. PMlR, 2019.
>
> [3] Smith, Damian, Harvey Mannering, and Antonia Marcu. "Functional Alignment Can Mislead: Examining Model Stitching." Forty-second International Conference on Machine Learning.
>
> [4] Balogh, András, and Márk Jelasity. "How not to stitch representations to measure similarity: Task loss matching versus direct matching." Proceedings of the AAAI Conference on Artificial Intelligence. Vol. 39. No. 15. 2025.
>
> [5] Balogh, András, and Márk Jelasity. "On the functional similarity of robust and non-robust neural representations." International Conference on Machine Learning. PMLR, 2023.
>
> [6] Csiszárik, Adrián, et al. "Similarity and matching of neural network representations." Advances in Neural Information Processing Systems 34 (2021): 5656-5668.
>
> [7] Nanda, Vedant, et al. "Measuring representational robustness of neural networks through shared invariances." International Conference on Machine Learning. PMLR, 2022.
>
> [8] Sarıyıldız, Mert Bülent, et al. "Fake it till you make it: Learning transferable representations from synthetic imagenet clones." Proceedings of the IEEE/CVF conference on computer vision and pattern recognition. 2023.

---

### Official Review · Reviewer_hyyX · 2025-11-01

**Soundness:** 3
**Presentation:** 2
**Contribution:** 2
**Rating:** 2
**Confidence:** 3

**Summary:**

This work proposes a new approach to model stitching to measure functional alignment to two different models. The proposed approach directly matches the representations for every layer after the stitching point. The authors perform a wide range of experiments that show this stitching approach is more sensitive that alternative methods for distinguishing models. The authors also look at the effect of stitching on a different dataset with slightly perturbed representations at the stitching layer.

**Strengths:**

The authors present an interesting conceptual argument for this new model stitching approach, motivated by previously identified weaknesses of existing stitching methods. They then go on to perform a series of experiments based on tests conducted in prior work.

**Weaknesses:**

While conceptually interesting, the results appear to show that direct matching (DM) gives effectively the same interpretation as the proposed FuLA method. This makes sense since FuLA would only differ from DM when the match is poor. It would seem that DM would then be the preferable method for its simplicity. FuLA also requires the two networks being compared to have the same architecture, which DM does not require.

Moreover, the results of the paper are poorly presented. Broadly speaking, the figures are confusing to interpret due to poor labeling, minimal captions, and the size of the text, which makes most of the results almost impossible to read on paper. For example, the titles of the plots in Fig. 4 are never defined or referenced elsewhere in the paper or in the caption. Fig. 6 is even more confusing, where the x-axis label seems to be important but is never explained.

**Questions:**

1. For the Identically Represented Inputs (IRI) datasets, why is it necessary to generate a completely new dataset if your goal is simply to perturb the output of the front model at the stitching layer? Why not avoid the whole optimization problem and just directly perturb/add noise to the representation at the stitching layer?
2. On line 328, why is the conclusion that the forward compatibility notion doesn't differentiate among different models when it clearly shows different behavior (the referenced "dip") in the plot? It seems to clearly differentiate it in this case.
3. Why use a completely different front model in section 3.2?
4. Why report rAuA only for the robustness examples and not the other examples? The way this metric is reported, via a label on a plot, is also poor and would be much easier to read in a table.

---

> ### Author Response · Authors · 2025-12-03
> **Responding to hyyX (1/4)**
>
> Dear hyyX, thank you for taking the time to review our work and provide valuable feedback. Please find our responses to the points you raised below.
>
> **Summary**
>
> Let us kindly start by addressing a point read in the summary which is a potential source of misunderstanding.
>
> > The authors also look at the effect of stitching on a different dataset with slightly perturbed representations at the stitching layer.
>
> At no point throughout the paper do we investigate such effects. Note that establishing stitching alignment under **arbitrarily perturbed representations** at the stitching layer does not constitute a principled design choice in the context of functional similarity evaluation. For example, any observed similarity deviation under varying levels of noise during stitching would only reflect the noise robustness of the stitching operation itself, rather than revealing any meaningful properties of functional similarity between models.
>
> Instead, we investigate the effect on stitching using Identically Represented Inputs (IRIs) (i.e., synthetically crafted data points that are represented identically to the actual data points at the stitch level of the *front mode*l). The rationale behind this is to probe for Requirement. B (Backward compatibility) **realizing the concept of backward compatibility within model stitching* as conceptually motivated in Sec.1 and detailed in Sec.2.3.1. This has been one of our main and arguably the one with the biggest impact, as also highlighted by the rest of the reviewers.

---

> ### Author Response · Authors · 2025-12-03
> **Responding to hyyX (2/4)**
>
> **Weaknesses**
>
> > W.1 the results appear to show that direct matching (DM) gives effectively the same interpretation as the proposed FuLA method …. DM would then be the preferable method for its simplicity.
>
> Through our experiments, we have established that: (i) the standard TLM, SLM, FuLA, and DM settings fail key sanity checks of functional similarity, (ii) their invariance-aware counterparts I-SLM, I-FuLA, and I-DM (i.e., the standard methods trained on IRIs) pass these tests, and (iii) among these, in the majority of cases, I-DM achieves the lowest similarity scores.
>
> In the presence of multiple valid stitching configurations (i.e., those that pass the sanity test), we should favor the one achieving the highest similarity score, otherwise, we would be **unnecessary discarding valid evidence** of higher functional alignment. Based on these, I-SLM or I-FuLA (depending on the architecture) are generally preferable to I-DM.
>
> > W.1 (cont.) FuLA also requires the two networks being compared to have the same architecture, which DM does not require.
>
> This is not the case, none of the stitching settings (i.e., TLM, SLM, FuLA or DM) require the compared/stitched networks to have identical architectures. Let us clarify the relationship between FuLA and DM as these are the focus of the statement.
>
> Under DM, the $T_\theta$ (i.e., the stitching layer) is optimized such that applying it to the *front model’s* output it produces features similar to those generated by the *end model* at the stitch level.
>
> Under FuLA, the $T_\theta$ is optimized such that applying it to the *front model’s* output it produces features that (i) are similar to those generated by the standalone *end model* at the stitch level (i.e., DM), and (ii) when processed by the *end model*, leads to feature representations similar to those generated by the standalone *end model* throughout all layers following the stitch level up to the penultimate layer.
>
> The relationship between DM and FuLA was also outlined in Sec.2.2 where the FuLA’s objective were formally defined (see Eq. 4).
>
> &nbsp;
>
> > W.2 Broadly speaking, the figures are confusing to interpret due to poor labeling, minimal captions, and the size of the text, which makes most of the results almost impossible to read on paper.
>
> We have updated the figures to improve readability based on the feedback received from all reviewers as well as made the caption more verbose.
>
> > W.2 (cont.) For example, the titles of the plots in Fig.4 are never defined or referenced elsewhere in the paper or in the caption.
>
> To improve clarity, we now explicitly introduce and reference the titles used in Fig.4 within the main text at the appropriate locations.
>
> > W.2 (cont.) Fig.6 is even more confusing, where the x-axis label seems to be important but is never explained.
>
> All stitching plots including Fig.4 and Fig.6 have the same x-axis label which denotes the depth of the front model composition (as described in the relevant paragraph in Sec.3).
>
> However, since Fig.4. and Fig.6. differ by their y-axis label we would also like to cover this angle, in case there was a typo during the statement’s formulation.
>
> The y-axis in Fig.4. refers to the agreement probability whereas the y-axis in Fig.6 refers to the excessive agreement probability (i.e., the difference between the agreement probability achieved in the shortcut dataset and the agreement probability achieved in the shortcut dataset with class-shifted shortcuts). Both of these were described in Sec.3 and Sec.3.2 respectively.

---

> ### Author Response · Authors · 2025-12-03
> **Responding to hyyX (3/4)**
>
> **Questions**
>
> > Q.1 For the Identically Represented Inputs (IRI) datasets, why is it necessary to generate a completely new dataset if your goal is simply to perturb the output of the front model at the stitching layer? Why not avoid the whole optimization problem and just directly perturb/add noise to the representation at the stitching layer?
>
> We believe this question stems from the misunderstanding earlier pointed out in the summary of our work. Our aim was not to evaluate functional similarity under arbitrary noise injected at the stitching layer. Instead, our goal was to jointly assess whether the *front* and *end models* are forward and backward compatible as per Reqs. A and B respectively.
> Constructing the IRIs dataset was necessary (and therefore the optimization problem generating it) because it provides data points that the *front model* perceives as similar to the regular data points at the stitch level, while the end model may not. This property can not be replicated by stitching on noise-induced perturbations of the front model’s output representations.
>
> &nbsp;
>
> >Q.2 On line 328, why is the conclusion that the forward compatibility notion doesn't differentiate among different models when it clearly shows different behavior (the referenced "dip") in the plot? It seems to clearly differentiate it in this case.
>
> For the fourth panel in Fig.4, indeed all forward compatibility settings display either a minor (i.e., TLM and SLM, FuLA) or a major (i.e., DM) dip in agreement probability by the mid layers. However, we argue that this constitutes **insufficient differentiation** since all of these settings ultimately recover to the agreement probability achieved by the standalone front model at the deeper layers.
>
> This recovery is problematic because the functional divergence between the deep layers of differently initialized models trained on regular data (second panel Fig.4) appears similar to the divergence between the model trained on regular data and that trained on shortcuts (fourth panel Fig.4). In other words, forward compatibility can not distinguish between deep layers responding to abstractions learned from regular data and those responding to shortcut cues, a pattern we consistently observe across our experiments.
>
> Under this interpretation, one would expect comparable transfer learning performance from the penultimate layers of either *front model* (either trained on regular or shortcut data). However, this is implausible since the model trained on the pixel shortcuts (CIFAR-10 w/ pixel shortcuts) has not learned any meaningful (i.e., generalizable) features as indicated by its random chance performance in the absence of shortcuts (see footnote 3).
>
> To improve the clarity and  precision of our claim, we replaced “models” with “model compositions”.
>
> &nbsp;
>
> > Q.3 Why use a completely different front model in section 3.2?
>
> In Sec.3.2, we investigate whether model stitching can leverage **previously unseen predictive information** to improve compatibility. This behavior is undesirable in the context of functional similarity evaluation , as it suggests that the stitching layer can engage in learning and therefore “defeats the aim of faithfully studying the original representations” [1]. Identifying a single configuration under which model stitching **consistently exhibits this behavior provides sufficient evidence that the undesirable behavior can occur**.
>
> We found that the effect was more pronounced both in terms of consistency and intensity, as measured by the excessive agreement, when stitching between models trained on different datasets (e.g., LS-ImageNet-10 and CIFAR-10). Therefore, we used cross-task stitching configurations in Sec.3.2 to illustrate the said undesirable behavior. Nevertheless, we note that the effect was also observed when stitching models trained on the same dataset, though less consistently and to a much smaller extent.
>
> To tighten the narrative, we included a brief explanation of why the cross-task configuration is used to demonstrate this undesirable behavior and added a reference indicating that the effect can also be observed, in the intra-task case (see Sec.3.2).

---

> ### Author Response · Authors · 2025-12-03
> **Responding to hyyX (4/4)**
>
> **Questions (cont.)**
>
> > Q.4 Why report rAuA only for the robustness examples and not the other examples?
>
> As discussed in Sec.3.3, the agreement probability adjusts the similarity relative to the one achieved by the standalone *front model* allowing for inter-configuration comparison where their standalone *front models* achieve significantly different agreement with respect to the *end model*.
>
> Sec.3.1 [Functional similarity across information variants]: The main argument used here is that when probing solely for forward compatibility the agreement curve is dominated by the agreement probability achieved by the standalone *front model* with respect to the *end model*.  As these *front models*, in the relevant comparisons, achieved comparable agreement with the *end models*, reporting the rAuA was not
> necessary.
>
> Sec.3.2 [Functional similarity under unseen predictive information]: In all stitching configuration, the agreement probability achieved by the standalone *front models* were comparable. Based on that, inter-configuration comparisons were trivial and similar to Sec 3.1, reporting the rAuA was not necessary.
>
> Sec.3.3 [functional similarity between robust and non-robust models]: In the various stitching configurations the *front models* achieved varying levels of agreement probability with respect to the *end models* e.g., third column in Fig.6. In that case, adjusting for the agreement of the standalone *front model* (i.e., via the rAuA metric defined in the same section) allows to infer that non-robust to robust models are roughly equally compatible with respect to both regular and adversarial samples. On the other hand,  through naive agreement aggregation, we would infer that these are significantly more compatible with respect to the regular samples.
>
> Based on these, we have chosen to report the rAuA only for Sec 3.3.
>
> > Q.4 (cont.) The way this metric is reported, via a label on a plot, is also poor and would be much easier to read in a table.
>
> Thank you for the suggestion. We agree that tables can improve readability in many cases; however, in this specific context we believe that separating the rAuA values from their corresponding stitching plots would reduce clarity (e.g., forcing the readers to cross-reference between stitching plots and tables introducing additional cognitive load).
> In our view, this integrated format effectively serves its purpose of distilling the stitching plot into a single quantitative metric.
>
> &nbsp;
>
> **References:**
>
> [1] Bansal, Yamini, Preetum Nakkiran, and Boaz Barak. "Revisiting model stitching to compare neural representations." Advances in neural information processing systems 34 (2021): 225-236.

---

### Author Response · Authors · 2025-12-04
**Summarizing the rebuttal**

We thank all reviewers for their thoughtful and constructive feedback, which has helped us improve the clarity of the presentation and expand the scope of our study. Below, we provide a brief summary of the key points of the rebuttal for the Area Chair’s consideration.

&nbsp;

 **1. Core concepts presentation (MHaa, HTsy, ZFEj)**:
- **Concern:** The concepts of forward and/or backward compatibility are not communicated early enough with sufficient clarity.
- **Response:** To address this, we included and discussed a toy functional alignment example in Sec.1 (Fig.1), demonstrating the impact of considering both forward and backward compatibility, compared to accounting for only forward compatibility.

&nbsp;

 **2. Result presentation (hyyX, MHaa, HTsy, ZFEj)**:
- **Concern:** The stitching plot design is hard to interpret, and the captions provide minimal explanation of the results.
- **Response:** We re-designed the stitching plots by removing the legends outside the figure and increasing the font size. Additionally, we included more detailed captions and extended the explanation of how these stitching plots should be interpreted.

&nbsp;

 **3. Contradicting convergent representation notion (MHaa, HTsy)**:
- **Concern:** There is literature supporting that indipendently trained networks converge to similarly structured latent space and therefore functional divergence is counterintuitive.
- **Response:** We included a brief discussion in Sec.3.4 relating our findings to the *seemingly* contradicting literature, ultimately argueing that representation convergence and functional divergence can co-exist.

&nbsp;

 **4. Limited experimental score (MHaa)**:
- **Concern:** Evidence is only provided for small-scale datasets and architectures, with missing comparisons to representation similarity baselines.
- **Response:** We extended the experimental scope by performing the analysis on the high-resolution ImageNet-100 dataset with ResNet-18 and ViT-Base architectures, verifying that the initial conclusions generalize and remain intact. Additionally, we compared our method with a widely used representation similarity metric (i.e., CKA), demonstrating that our approach yields unique insights.

&nbsp;

 **Final remark**: The above provides a brief summary of the main axes of critique and our responses. We reiterate that all reviewers raised relevant points, which have been higgly valuable in improving our work as we addressed them.

Thank you in advance for reviewing the rebuttal.

---

### Meta-Review · Area_Chair_UHBG · 2025-12-19

**Summary:**

This paper looks at model stitiching, in particular in the setting where the original models are trained on different “information cues”. The original reviewed highlighted a number of challenges with the submission:

- Clarity and presentation. Multiple reviewers noted that the paper is difficult to read, with several concepts that crucial to the setting yet not properly defined. The authors have made several edits to the paper that partially clarifies these points. Overall, I think that the new version has improved on these points which would marginally raise some score. Still, several concepts remain poorly discussed or defined. Also the new paragraph on “A note on convergent representations” is not very clear to me. It’s in response to reviewer’s HTsy question on the paper’s premise and position to the literature, but the new language simply acknowledges the divergence in the literature, without giving an autoritative answer.

- Limited experiments. Several reviewers complained about the experimental scope both in terms of data sets and models. The authors have addressed this by adding a high-resolution imagenet 100 experiment and a ViT Base. This part was less convincing for me, because the results in the main paper are still only about ResNets. The ViT experiments are in the appendix and it was difficult for me to follow the argument that the results are consistent with the ResNet ones (e.g., Figure 10 seems to indicate major differences across architectures). Overall, it’s hard for me to judge whether the reviewers will have changed their scores significantly in response to this point.

- Reviewer ZFEj raised a question about whether DM is essentially equivalent to FuLA except in corner cases. The author’s reply was not satisfactory, as it seems to confirm the point.

Overall, even in normal circumstances, I would have advised the reviewers that the paper should undergo review again. The new experiments and the narrative need to be cohesively incorporated and right now they are not, in my opinion.

**Reviewer Concerns:**

See above. The reviewers raised multiple concerns, largely about presentation and experiment. I think the authors partially addressed them and the paper can be considered borderline now. Still, it's a substantial change and even in normal circumstances, I'd advise another round of review.

**Reviewer Scores:**

I think the reviewers may have marginally increased their scores but that would still likely not move the paper in the accept pile.

---

### Decision · Program_Chairs · 2026-01-26

Reject